

**Mapping Global Non-Floodplain Wetlands**
Charles R. Lane[1], Ellen D'Amico[2], Jay R. Christensen[3,★], Heather E. Golden[3,★], Qiusheng Wu[4], and
Adnan Rajib[5]
[1] U.S. Environmental Protection Agency, Office of Research and Development, Center for Environmental

7        Measurement and Modeling, Athens, Georgia, United States of America

[2] Pegasus Corporation c/o U.S. Environmental Protection Agency, Office of Research and Development,

9        Cincinnati, Ohio, United States of America

[3] U.S. Environmental Protection Agency, Office of Research and Development, Center for Environmental

11       Measurement and Modeling, Cincinnati, Ohio, United States of America

[4] Department of Geography & Sustainability, University of Tennessee, Knoxville, Tennessee, United

13       States of America

[5] Hydrology and Hydroinformatics Innovation Lab, Department of Environmental Engineering, Texas

15       A&M University, Kingsville, Texas, United States of America

★ These authors contributed equally to this work
**Correspondence:** Charles Lane (lane.charles@epa.gov) and Ellen D'Amico (damico.ellen@epa.gov)
**Abstract.** Non-floodplain wetlands – those located outside the floodplains – have emerged as integral
components to watershed resilience, contributing hydrologic and biogeochemical functions affecting
watershed-scale flooding extent, drought magnitude, and water-quality maintenance. However, the
absence of a global dataset of non-floodplain wetlands limits their necessary incorporation into water
quality and quantity management decisions and affects wetland-focused wildlife habitat conservation



outcomes. We addressed this critical need by developing a publicly available Global NFW (non-
floodplain wetland) dataset, comprised of a global river-floodplain map at 90 m resolution coupled with a
global ensemble wetland map incorporating multiple wetland-focused data layers. The floodplain,
wetland, and non-floodplain wetland spatial data developed here were successfully validated within 21
large and heterogenous basins across the conterminous United States. We identified nearly 33 million
potential non-floodplain wetlands with an estimated global extent of over 16 million km$^2$. Non-floodplain
wetland pixels comprised 53% of globally identified wetland pixels, meaning the majority of the globe's
wetlands likely occur external to river floodplains and coastal habitats. The identified Global NFWs were
typically small (median 0.039 km$^2$), with a global median size ranging from 0.018-0.138 km$^2$. This novel
geospatial Global NFW dataset advances wetland conservation and resource-management goals while
providing a foundation for global non-floodplain wetland functional assessments, facilitating non-
floodplain wetland inclusion in hydrological, biogeochemical, and biological model development. The
data are freely available through the United States Environmental Protection Agency's Environmental
Dataset Gateway (https://gaftp.epa.gov/EPADataCommons/ORD/Global_NonFloodplain_Wetlands/) and
through https://doi.org/10.23719/1528331 (Lane et al., 2023).
**1      Introduction**
Wetlands are recognized as globally important ecosystems providing functions leading to critical
provisioning (e.g., food, fresh water for domestic, agricultural, and industrial use) and regulating services
(e.g., flood and drought mitigation, water purification and waste treatment, and habitat; Millennium
Ecosystem Assessment, 2005). Despite their functional importance, wetlands are threatened worldwide by
myriad anthropogenic disturbances, including sea-level rise (IPCC, 2014), drainage and filling (Davidson
et al., 2014), water abstraction (Liu et al., 2017), consolidation (McCauley et al., 2015), invasive species
(Zedler and Kercher, 2004), and changing precipitation and temperature patterns (Winter, 2000). These
widespread and globally prevalent alterations to wetlands affect their functioning, resulting in increased



downgradient flooding (Golden et al., 2021), modified stream baseflows (Buttle, 2018), reduced pollution
mitigation (Evenson et al., 2018a), and habitat loss (Uden et al., 2015).
Watershed-scale wetland management is currently hampered by the paucity of accurate and fine-grained
maps of wetland location (Creed et al., 2017; Christensen et al., 2022). However, methods to identify
existing aquatic systems, including wetlands, that provide functions at global scales have recently
emerged, such as the Landsat-based 30 m global surface-water inundation data (Pekel et al., 2016), finer-
resolution satellite-based landcover maps (e.g., Zanaga et al., 2021), and groundwater-driven aquatic
system characterizations (Fan et al., 2013). In addition, methods utilizing digital elevation models to
identify topographic depressions likely to support aquatic systems with characteristic wetland features,
such as saturated soils and/or ponded waters, have also regionally proliferated (Wu et al., 2019a; Wu et
al., 2019b; Christensen et al., 2022).

These advancements in mapping wetland location, such as those located within the river floodplain or
geographically distal from floodplains, allow resource managers to better incorporate wetland
biogeochemical, hydrological, and biological functions and concomitantly ecosystem services into their
decision-making efforts. For instance, incorporating *floodplain* wetlands into decision-making advances
the wise management and conservation of mapped riparian ecosystems (Tullos, 2018; Kundzewicz et al.,
2018). Thus, recognizing the importance of wetlands located within active river floodplains, land-
management decisions are being made to quantify the functions and ecosystem services of these wetlands
and incorporate them into watershed-scale hydro-ecological decisions (e.g., Makungu and Hughes, 2021;
Rajib et al., 2021).

However, *non-floodplain wetlands* are typically not incorporated into watershed-scale conservation and
management planning (e.g., Sullivan et al., 2019), thereby ignoring their contributions to watershed-scale
resilience in response to biogeochemical and hydrological disturbances (Rains et al., 2016; Golden et al.,



2021; Lane et al., 2022). Non-floodplain wetlands are abundant inland wetlands located distally from the
floodplains of rivers and lakes (Lane and D'Amico, 2016; Lane et al., 2018). Though typically small
(Cohen et al., 2016), high biogeochemical processing rates within non-floodplain wetlands have resulted
in these systems being termed bioreactors (Marton et al., 2015). Indeed, a literature review of over 600
articles found that the highest reactivity rates (pollutant mass removal per unit time) were found in the
smallest water bodies and wetlands (Cheng and Basu, 2017). Further, the high reactivity of individual
non-floodplain wetlands can cumulatively improve downgradient water quality conditions (Golden et al.,
2019; Evenson et al., 2021). Non-floodplain wetlands may therefore have an outsized impact on a
watershed's water quality.

Non-floodplain wetlands are also important ecosystems affecting water quantity (i.e., for storing and
gradually releasing water to downgradient rivers and streams). Specifically, precipitation is captured and
stored in non-floodplain wetlands prior to being discharged downgradient. During this storage period,
water can infiltrate to recharge aquifers, evaporate or transpire, or eventually "spill" overland and be
transported downstream (Jones et al., 2018; Buttle, 2018). These non-floodplain wetland water storage
functions attenuate storm flows (Shaw et al., 2012; Fossey and Rousseau, 2016; Blanchette et al., 2022)
and recharge groundwaters (Bam et al., 2020), thereby mitigating flood-hazards (Mclaughlin et al., 2014)
and ameliorating drought conditions by maintaining baseflow (Ameli and Creed, 2019).

Despite the important functions provided by non-floodplain wetlands (Biggs et al., 2017; Chen et al.,
2022) a substantive data gap remains: no global maps or datasets exist identifying the geospatial location
of non-floodplain wetlands and open waters. Regionally focused efforts, such as the recent work by Lane
and D'Amico (2016) and Lane et al. (2022) mapped the extent of non-floodplain wetlands (also known as
geographically isolated wetlands, Leibowitz, 2015; Mushet et al., 2015) across the geospatially data-rich
conterminous United States (CONUS, see abbreviation list Appendix A). They found that 16-23 % of



freshwater systems were potential non-floodplain wetlands, suggesting a substantial yet hitherto unknown
portion of the globe's wetlands are likely also this vulnerable water resource.

Fortunately, geospatial data for identifying aquatic systems, including wetlands, are burgeoning (Khare et
al., in review). Global land cover and land use geospatial datasets that include a wetland cover class
continue to propagate (Hu et al., 2017a), taking advantage of both lengthy time-series Landsat data
(Homer et al., 2020) as well as recently launched advanced high-resolution and/or synthetic aperture radar
(SAR) equipped satellites (e.g., Sentinel-1, Sentinel-2, plus many commercially available platforms;
Martinis et al., 2022) and topographic data sources and analyses (e.g., Wu et al., 2019b). Examples
include the GlobeLand30 (Chen et al., 2015), the European Space Agency (ESA) WorldCover 2020
(ESA, 2020), the Dynamic World (Brown et al., 2022), as well as consortiums focusing on annual land
cover change mapping (e.g., Tsendbazar et al., 2021).

Lehner and Döll (2004) were amongst the first to publish a geospatially explicit global map focusing on
wetland extents. Their Global Lakes and Wetlands Database provides 1 km estimates of wetland
abundance. More recent and/or higher resolution wetland-focused datasets have emerged, including the 1
km global dataset from Hu et al. (2017b) that incorporates precipitation and a topographic wetness index,
and the multi-sourced 500 m composite maps of regularly flooded and groundwater-driven wetlands by
Tootchi et al. (2019). Tootchi et al.'s (2019) approach identified small and scattered wetlands. However,
they recognized the limitations inherent in their global product (ca. 500 m per pixel resolution) resulted in
omission errors for many wetland systems, especially those smaller than their 500 x 500 m (25 ha) data
resolution. This suggests, and Tootchi et al. (2019) acknowledged, that many (non-floodplain) wetlands
were omitted in the Tootchi et al. (2019) 500 m global product. Cohen et al. (2016) determined non-
floodplain wetlands in the CONUS are "unambiguously small", e.g., their average non-floodplain wetland
area was just over two hectares (2.1 ha). Based on the "all or nothing" methodological approach in
Tootchi et al. (2019), > 12.5 ha of a given 25.0 ha [one homogenous pixel] cell would have to be



identified as wetland in their resampling of the finer-scale data – much larger than the average 2.1 ha
wetlands found in Cohen et al. (2016).

Concurrent with increasingly available global land cover and wetland data, there is an increasing global
focus on deriving floodplain and flood-prone areal extents within river networks based on high-resolution
topographic data and hydraulic modeling (Tullos, 2018; Kundzewicz et al., 2018; Rajib et al., in review).
For instance, Sampson et al. (2015) created a global 90 m map of flood-prone areas between $60^{\circ}$ N and
$56^{\circ}$ S. Nardi et al. (2019) developed a global floodplain dataset at 250 m resolution that extended from $60^{\circ}$
N to $60^{\circ}$ S, based on geomorphic or terrain-based analyses of floodplain elevations. The evolution of the
MERIT Hydro 90 m global hydrography dataset by Yamazaki et al. (2019) has created additional
opportunities to further advance the derivation of global floodplains, with improved identification of flow
accumulation area, river-basin shape, and river channel location.

These wetland-location and floodplain-extent data are critical for watershed-scale sustainable aquatic
resource policy decisions (Creed et al., 2017; Golden et al., 2017). The lack of these data can result in
disproportionately large model errors and potentially misguided management decisions when non-
floodplain wetlands are not incorporated in hydrological and biogeochemical models, ignoring their
watershed-scale impacts on flooding, drought, and water quality (Evenson et al., 2018a; Rajib et al., 2020;
Golden et al., 2021).

Here, we provide the first global geospatial dataset of non-floodplain wetlands. We incorporate the recent
development of a high-resolution global floodplain mapping algorithm based on digital terrain models by
Nardi et al. (2019). We couple these spatial floodplain data with higher-resolution modifications to the
gridded global wetland and open water data layers developed by Tootchi et al. (2019) that incorporate the
Pekel et al. (2016) satellite-based inundation product, modeled groundwater-driven wetland extent (Fan et
al. (2013), and ancillary satellite landcover data from Herold et al. (2015). We test the applicability of our



global dataset of non-floodplain wetlands in 21 large and spatial-data rich watersheds across the CONUS.
This novel global product identifying non-floodplain wetlands provides for the quantification and
estimation of the locations and extent of important aquatic systems with abundant hydrological,
biogeochemical, and biological functions, filling a noted research gap while delivering useful data for
informed natural resource decision-making and management (Creed et al., 2017; Lane et al., 2022).

**2       Methodology and data**

Identifying global non-floodplain wetlands required the following steps: 1) determination of global
floodplain extent, 2) identification of the global distribution of wetlands, 3) spatial overlay (masking) of
floodplains and wetlands to derive a non-floodplain wetland data layer, and 4) data verification and
accuracy assessment. Steps 1-3 are outlined in a flow chart given in Figure 1.

**2.1      Global floodplain data**

Nardi et al. (2019) combined space-borne elevation data and terrain analysis with a novel open-source
algorithm to delineate the geomorphic floodplains across the globe between 60º N and 60º S latitudes.
Conceptually, Nardi et al. (2019) identified floodplains from surrounding hillslopes as those low-lying
landscape features that have been naturally shaped by accumulated geomorphic effects of past flood
events. The original Nardi et al. (2019) dataset was limited in its spatial extent (60º N-60º S) and
resolution (250 m); this study sought to delineate global floodplain extent while concurrently identifying
floodplain features further up the river network than possible with 250 m pixels. Hence, we utilized the
freely available Nardi et al. (2019) GFPlain v1.0 algorithm and coupled this with the MERIT Hydro
(Multi-Error Removed Improved Terrain, Yamazaki et al., 2019), global raster digital terrain model data
to develop a higher resolution (90 m) geomorphic riverine floodplain for the globe, termed hereafter
GFPlain90.




**Figure 1.** Data flow chart identifying the main data sets and processes involved in deriving the Global Floodplain
and Global Wetland data layers, as well as the intersection of those data to create the Global Non-floodplain
Wetlands data product. Curved boxes represent final products, and abbreviations may be found in the text and
Appendix A.



The development of GFPlain90 required multiple steps. We first extracted elevation data from MERIT
Hydro, reprojected the data in UTM zones to prevent distortion when using the GFPlain algorithm, and
then developed the drainage network, drainage area, flow accumulation and flow direction data from
these data using the scaling parameters in Nardi et al. (2019). We established 20 km$^2$ as the minimum
contributing-area threshold required to create the drainage network, balancing the development of a
global stream-network distribution and extent with computational requirements. We then globally
organized the data by HydroBASINS Level 4 basins (Lehner and Grill, 2013). HydroBASINS provides
seamless watershed boundaries and subbasin delineations at global scales; there are 1,342 Level 4
HydroBASINS globally. The floodplain extent resolution of GFPlain90 was resampled (using nearest
neighbor) to 30 m for subsequent performance assessment and overlap analyses with the wetland spatial
data. All spatial analyses in this study were conducted using ArcGIS Pro v.2.9.x (ESRI, Redlands,
California) and GRASS GIS v 7.4.4 (OSGEO, Beaverton, Oregon).

**2.2    Global Wetland data**

Tootchi et al. (2019) developed a widely used composite global wetland map at ~500 m by combining
multiple data sources, including both satellite-based surface-water inundation mapping and vegetation
classification coupled with model-based approaches capturing important groundwater-driven wetland
systems. We specifically used the Tootchi et al. (2019) composite map consisting of both regularly
surface-water flooded and groundwater discharge-maintained wetlands as the foundation for our global
wetland map.

**2.2.1    Original composite wetland data**

Regularly flooded wetlands (RFWs) derived by Tootchi et al. (2019) were based on three data sources: 30
m resolution Global Surface Water (GSW) by Pekel et al. (2016), 300 m Climate Change Initiative (CCI)



land cover data by Herold et al. (2015), and 500 m GIEMS-D15 wetland extent data by Fluet-Chouinard
et al. (2015). GSW data used by Tootchi et al. (2019) were developed from Landsat satellite imagery
analyses of pixels identified as inundated at least once during the 32-year period of record by Pekel et al.
(2016). CCI input wetland data for Tootchi et al. (2019) included both inundated and wetland vegetation-
classed pixels assessed during the period 2008-2012 by Herold et al. (2015). For GIEMS-D15, data
included were the mean annual maximum extent of pixels identified as wetlands using multi-sensor
satellite data by Prigent et al. (2007), downscaled to ~500 m resolution by Fluet-Chouinard et al. (2015).
GSW and CCI input data were resampled to ~500 m resolution using an "all or nothing" approach by
Tootchi et al. (2019). This means that a pixel categorization of "wetland" at 500 m resolution was given
by Tootchi et al. (2019) only if the majority of resampled finer-resolution input pixels were classed as
wetlands. The resampling from 30 m and 300 m to 500 m resulted in a loss of informative spatial data on
wetland extent from GSW and CCI. Tootchi et al. (2019) calculated that RFWs cover approximately 9.7
% of the global land area (excluding lakes [sourced from (Messager et al., 2016)], Antarctica, and the
Greenland ice sheet).

Groundwater-driven wetlands (GDWs in the analysis of Tootchi et al., 2019) used in this study were
based on the water-table depth estimates by Fan et al. (2013). Fan et al. (2013) developed a 1 km
resolution groundwater map based on climate and terrain variables that was validated by over 1 million
government-recorded and published observations. Fan et al. (2013) estimated that shallow groundwater
influenced nearly 15 % of groundwater-fed surface features, explaining important wetland patterning at
global scales (as well as vegetation classes at local and regional scales). A water-table depth threshold of
≤ 20 cm was used by Tootchi et al. (2019) to identify groundwater-driven wetlands and they resampled
these data to ~500 m cell resolution. The GDW distribution based on water table depths covered
approximately 15 % of the global land mass (including large portions of the Amazon basin, coastal zones,
and North American and Siberian peatlands).




Tootchi et al. (2019) merged the RFW and GDW maps to form a union product with a high correlation
with available evaluation data, which they called the composite wetland-water table depth (hereafter CW-
WTD). They measured an approximately 3.8 % overlap between the total land pixels identified as
wetlands in both the RFW and GDW maps that comprise the CW-WTD, suggesting the different input
maps capture different wetland types. At the global scale, Tootchi et al. (2019) reported spatial Pearson
correlations between CW-WTD (wetland fractions at 3 arcmin, or ~4.9 km grids) and wetlands within
GLWD (Lehner and Döll, 2004) and Hu et al. (2017b) as r=0.34 and r=0.43, respectively. Tootchi et al.
(2019, their Table 5 and S1) provided additional analysis of the correlations between their global wetland
product and existing benchmark data. The total CW-WTD global wetland estimate was ~ 21.1 % of the
land mass, or approximately 27.5 million km$^2$ (excluding large lakes, Antarctica, and the Greenland ice
sheet; Tootchi et al., 2019).

**2.2.2    Derived global wetland data**

To account for the acknowledged limitations of the Tootchi et al. (2019) data and to accurately identify
more of the existing small and, specifically, non-floodplain wetlands across the globe (e.g., those <25 ha),
we improved upon and augmented the CW-WTD (Tootchi et al., 2019) global wetland data layer with the
30 m native-resolution GSW (Pekel et al., 2016) and 300 m native-resolution CCI (Herold et al., 2015)
data. The inclusive wetland categories of Tootchi et al. (2019) were maintained, namely at least one
inundation event over a 32 year range (for GSW data) and CCI pixels defined as "…mixed classes of
flooded areas with tree covers, shrubs, or herbaceous covers plus inland water bodies…" (Tootchi et al.,
2019, p. 193). However, for our analysis we resampled the 500 m CW-WTD product to 30 m using the
nearest-neighbor approach and then added any identified wetland pixel from the CCI data (resampled
from 300 m to 30 m) and inundated pixel from the GSW data (30 m resolution). Resampling to a finer
resolution does not result in a loss of any data whereas resampling from a finer resolution to a coarser



resolution results in the loss of any data smaller than the chosen resolution. This resulted in a novel and
encompassing wetland ensemble end-product, hereafter termed the Global Wetlands dataset. This new
dataset is inclusive of both finer-resolution (30 m) data, thereby accounting for a wide range of wetland
sizes – such as smaller non-floodplain wetlands (Cohen et al., 2016) – that remained unmapped by
Tootchi et al. (2019).

**2.3      Global Non-Floodplain Wetlands (Global NFWs)**

To identify non-floodplain wetlands specifically, we overlaid our GFPlain90 floodplain data with our
mapped Global Wetlands data to mask wetland pixels collocated on the floodplain. Then, to avoid tidally
influenced wetlands, we conducted a region-group analysis to identify connected pixels abutting coastal
shorelines in order to mask wetlands in coastal areas (e.g., those directly abutting the shoreline and
spatially connected to tidally influenced areas). We used a four-directional contagion criterion to identify
connected pixels (i.e., those connected in cardinal directions). Subsequently, we applied a 1 km buffer to
the HydroBASINS (Lehner and Grill, 2013) coastline area and removed from our analyses any wetland
region-group partially or completely overlain by the 1 km coastline buffer. In addition, Tootchi et al.
(2019) removed lake systems ($\geq$ 10 ha) from their wetland-focused data by masking aquatic layers using
HydroLAKES (Messager et al., 2016). To avoid including large lakes in our emerging non-floodplain
wetland geospatial data, we also applied the HydroLAKES mask and removed lake systems $\geq$ 10 ha
(Messager et al., 2016) from our Global Wetlands dataset. Thus, our final global non-floodplain wetland
data product (hereafter Global NFWs) did not include fluvial floodplain wetlands nor coastal wetland
complexes and large open water lacustrine (lake-like, Cowardin et al., 1979) systems.

**2.4      Data verification and assessment**



We evaluated the global products developed here through comparison of high-resolution floodplain and
wetland extent data from 21 basins representing disparate climatic (according to the Köppen-Geiger
classification, Beck et al., 2018), elevation, and land-use gradients within the CONUS (Fig. 2;
summarized in Table B1). We specifically focused on the CONUS for product assessment because of its
wide-ranging data availability and diversity of physiographic and climatic regions.

**2.4.1   Verifying floodplain extent**

We used a recently developed machine learning (ML)-based 30 m resolution CONUS floodplain dataset
(Woznicki et al., 2019) as the benchmark to evaluate our GFPlain90 global floodplain data. Specifically,
the ML model by Woznicki et al. (2019) used the U.S. Federal Emergency Management Agency (FEMA)
100 yr floodplain (i.e., a 1 % chance of coastal or fluvial flood-inundation in a given year; Jakubínský et
al., 2021) as the training data, and subsequently used soil and topographic characteristic along with land
cover to identify potential floodplain grid cells across CONUS at 30 m resolution. Woznicki et al. (2019)
reported that their ML approach correctly identified ~79 % of the FEMA 100 yr coastal and fluvial
floodplains, providing spatially complete 100 yr floodplain coverage totaling 980,450 $km^2$ across the
CONUS.

**2.4.2   Verifying wetland and non-floodplain wetland extent**

We evaluated our inclusive Global Wetlands and Global NFWs datasets in 21 basins covering ~680,000
$km^2$ (Fig. 2). We contrasted our products to the 2016 National Land Cover Database (NLCD, Dewitz,
2019). The NLCD is a 30 m Landsat satellite-based geospatial product with an overall accuracy of 86 %
that incorporates high-resolution aerial imagery of wetland location for model parameterization and
calibration (Jin et al., 2019; Wickham et al., 2021). Three NLCD classes were selected for comparison
with the Global Wetland product: woody wetlands, emergent herbaceous wetlands, and open water. To
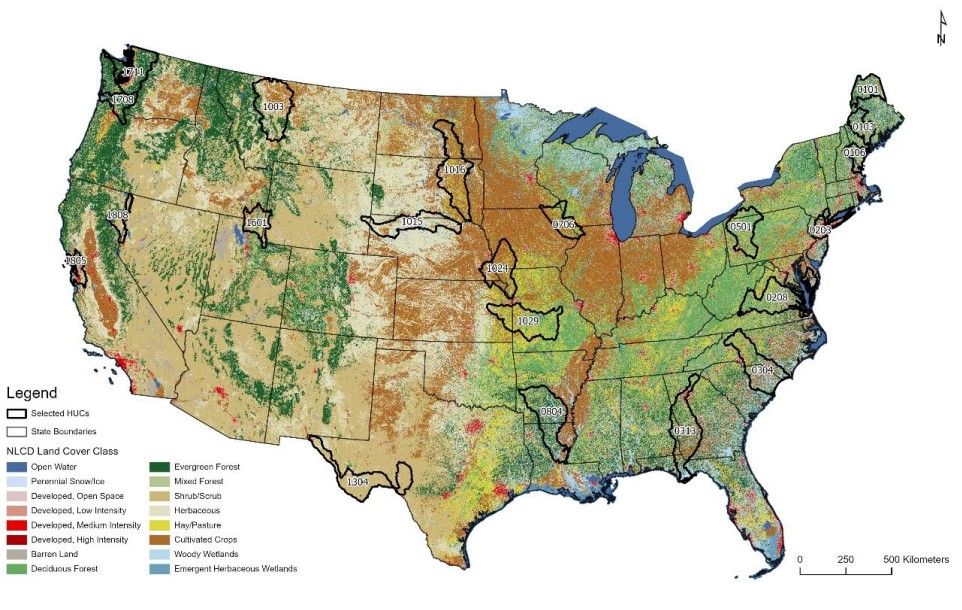


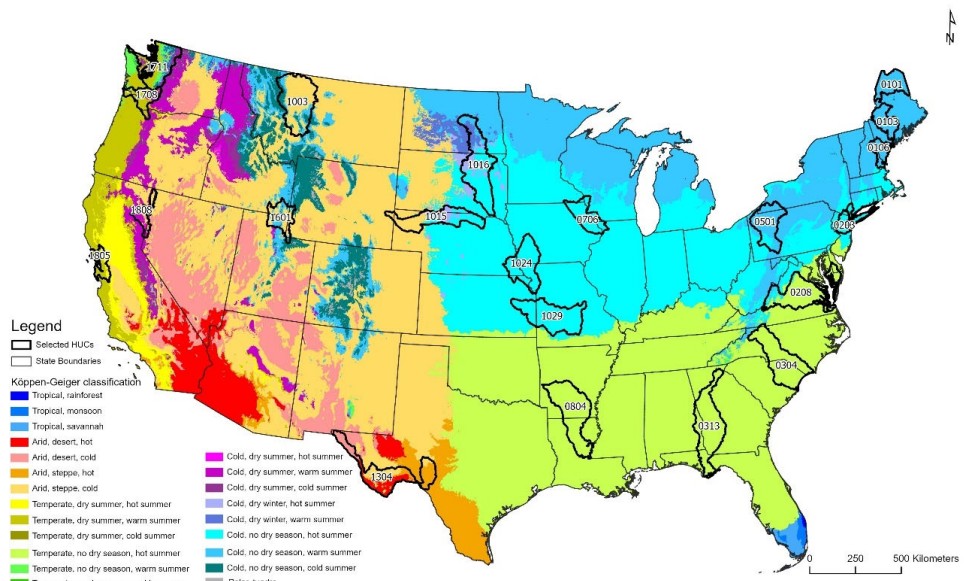


**Figure 2.** Twenty-one validation watersheds were selected from across CONUS to capture the breadth and extent of

land use (top) and climate and physiographic regions (bottom) within CONUS according to the Köppen-Geiger

classification (Beck et al., 2018); also summarized in Table B1). Land cover data are from NLCD (2019) and the

Hydrologic Unit Code (HUC) classifications are sourced from USGS Watershed Boundary Dataset (2022).



assess the relative improvement of our 30 m Global Wetlands and Global NFWs dataset with the 500 m
Tootchi et al. (2019) data, we also contrasted the CW-WTD with the NLCD classes within the
verification watersheds. For equal comparisons, following Tootchi et al. (2019) we used the Messager et
al. (2016) HydroLAKES to mask out large lake systems ($\geq$ 10 ha) from both the Global Wetlands and the
NLCD data within the 21 verification watersheds.

**2.4.3    Standard performance measures**

We evaluated the floodplain and wetland spatial data within the 21 validation watersheds using
commonly employed performance measures. Following Wing et al. (2017), we first created a contingency
table for our performance assessment (Table 1). As noted, we selected 20 $km^2$ as the minimum
contributing area to develop stream networks in our global floodplain analysis, a reasonable area for flow-
accumulation that balances computational efficiency for global geospatial model development. Woznicki
et al. (2019), our benchmark floodplain dataset, used a 4.5 $km^2$ contributing area in their high-resolution
CONUS analysis. To appropriately compare between datasets of two varying resolutions, we removed
stream and river network components from the Woznicki et al. (2019) validation dataset developed with
contributing areas <20 $km^2$, as our model did not discern landscape data at that granularity.

**Table 1.** Contingency table of possible outcomes for each cell used in assessing the performance of either the
floodplain or wetland geospatially modeled data. We contrasted published benchmark data from Woznicki et al.
(2019) for floodplain extent against modeled GFPlain90 data. Wetland comparisons contrasted NLCD wetlands
(Dewitz, 2019, open water and wetland classes) against both Global Wetlands and Global NFWs data.

|  | Floodplain [or Wetland] in Benchmark data | Not Floodplain [or Wetland] in Benchmark data |
|---|---|---|
| **Floodplain [or Wetland] in Modeled data** | $M_1B_1$ | $M_1B_0$ |
| **Not Floodplain [or Wetland] in Modeled data** | $M_0B_1$ | $M_0B_0$ |




To provide a full assessment of our geospatial modeling performance, we contrasted our GFPlain90
floodplain dataset across the 21 validation watersheds using the approaches described below following
Sampson et al. (2015), Wing et al. (2017), and others (e.g., Bates and De Roo, 2000; Alfieri et al., 2014;
Sangwan and Merwade, 2015; Jafarzadegan et al., 2018; Woznicki et al., 2019). We first contrasted our
GFPlain90 floodplains to Woznicki et al. (2019), our benchmark floodplain data. We then analyzed the
watershed-scale comparison of our Global Wetlands product versus the NLCD wetlands (combined open
water and wetland classes), our benchmark wetlands data. We followed with a comparison focusing only
on our Global NFWs data and those NLCD wetlands and open water pixels that were determined to be
non-floodplain systems (i.e., NLCD data that also do not overlap the GFPlain90 data nor coastal waters
and with lakes >10 ha removed). These NLCD wetlands were our benchmark non-floodplain wetland
data. Lastly, we assessed the mean and aggregate error bias of our analyses by exploring results at coarser
spatial granularity (i.e., 1 km pixel size) along the riverine network (for floodplain assessment) and, for
wetland metrics, throughout the entirety of our 21 performance assessment watersheds (Sampson et al.,
2015; Wing et al., 2017). The metrics described below were used in our analyses.

*Hit Rate* (Bates and De Roo, 2000; Horritt and Bates, 2002; Tayefi et al., 2007; Alfieri et al., 2014;
Sampson et al., 2015; Wing et al., 2017; Jafarzadegan et al., 2018) also referred to as Recall (Woznicki et
al., 2019) and Correct (Sangwan and Merwade, 2015), measures how well a geospatial model
classification replicates the benchmark data but does not penalize for overprediction. *H* varies from 0,
where there is no overlap between the modeled data and the benchmark data, to 1 where the modeled data
completely contain the benchmark data.
$$Hit\ Rate\ (H) = \frac{M_1 B_1}{M_1 B_1 + M_0 B_1} \qquad (1)$$

*Precision* (Woznicki et al., 2019), also known as Spatial Coincidence (Tootchi et al., 2019), indicates the
proportion of the benchmark data that are correctly predicted and mapped in the modeled data. This
metric, *P*, also ranges from 0 to 1 with higher values indicating better performance.



$$Precision\ (P) = \frac{M_1B_1}{M_1B_1+M_1B_0} \tag{2}$$


The *False Alarm Ratio* ( Sampson et al., 2015; Wing et al., 2017) also known as the False Discovery
Ratio, quantifies modeled data overprediction relative to the benchmark data. *F* varies from 0 (zero false
alarms) to 1 (all false alarms); lower values are considered better performance. The False Alarm Ratio can
also be calculated as 1 - *Precision* (Woznicki et al., 2019).
$$False\ Alarm\ Ratio\ (FA) = \frac{M_1B_0}{M_1B_0+M_1B_1} \tag{3}$$


The *Critical Success Index* (CSI,  Bates and De Roo, 2000; Aronica et al., 2002; Werner et al., 2005;
Fewtrell et al., 2008; Alfieri et al., 2014; Sampson et al., 2015; Wing et al., 2017), also known as
Jaccard's Index (Tootchi et al., 2019), and Fit (Sangwan and Merwade, 2015), penalizes for both over-
and under-prediction, ranging from 0 (no match) to 1 (perfect match).
$$Critical\ Success\ Index\ (CSI) = \frac{M_1B_1}{M_1B_1+M_0B_1+M_1B_0} \tag{4}$$


Woznicki et al. (2019) utilized a performance metric, *F1*, which combines the *Hit Rate* (called Recall by
Woznicki et al. 2019) and *Precision* using their harmonic mean. *F1* also varies from 0 to 1, with higher
values indicating better performance.
$$F1 = 2\left(\frac{H \times P}{H + P}\right) \tag{5}$$


*Error Bias* (*EB*) characterizes the tendency of the model towards under- or over-prediction (Sampson et
al., 2015). Values of 1 indicate no bias, $0 \leq EB < 1$ indicates underprediction whereas $1 < EB \leq \infty$
indicates the model is tending towards overprediction.
$$Error\ Bias\ (EB) = \frac{M_1B_0}{M_0B_1} \tag{6}$$


Lastly, two additional metrics were calculated that assessed performance at the 30 arc-sec (~1 km) scale.
These measures, *Mean Absolute Error* and *Aggregate Error Bias* (Sampson et al., 2015; Wing et al.,
2017), characterize the data accuracy across large spatial extents. Large spatial extents are areas where 30
m data and overlap accuracy is less a concern than general dataset performance for broad-scale end-user
applications (e.g., when coarser, watershed-scale "lumped" hydrologic characterizations of water storage
are all that is required). For these metrics, both estimated and benchmark data were resampled to 1 km
resolution across the whole of each watershed; values within each 1 km pixel ranged from 0 to 1 and
represented the fraction of the 30 m resolution estimates and benchmark data. We assessed floodplain
estimates after calculating the fractional abundance comprising each 1 km$^2$ pixel within a 1 km buffer
around the Woznicki et al. (2019) floodplain data. We additionally analyzed all wetlands at the
watershed-scale as well as focusing on non-floodplain wetlands (e.g., wetlands exclusive of the
GFPlain90 floodplain or coastal connections, our target aquatic system).
$$\textit{Mean Absolute Error } (E_A) = \frac{\sum_{i=1}^{N}|M-B|}{N} \tag{7}$$

$$\textit{Aggregate Error Bias } (B_A) = \frac{\sum_{i=1}^{N}M-B}{N} \tag{8}$$

Where $M$ is the area estimated as floodplain (or wetland), $B$ is the benchmark floodplain (or wetland)
area, and $N$ is the number of 1 km cells with data. *Mean Absolute Error* and *Aggregate Error Bias* were
calculated for each of the 21 HUCs, following Wing et al. (2017).






**3      Results**

**3.1      Floodplain data performance**

The GFPlain90 floodplain data (Fig. 3) performed well when contrasted with the 100 yr coastal and
fluvial floodplain extent data from Woznicki et al. (2019), even though our analyses do not map coastal
floodplains. A median Hit Rate of 0.77 suggests that nearly 80% of the benchmark floodplain from
Woznicki et al. (2019) was similarly captured by the GFPlain90 floodplain data (Table 2). In addition, the
median False Alarm of 0.26 indicates that for every three pixels correctly identified as within the
Woznicki et al. (2019) floodplain, one pixel was incorrectly identified as such (i.e., a commission error
measure); this is evident in wider GFPlain90 floodplains in lower river reaches than predicted by
Woznicki et al. (2019). These performance values are similar to those reported by Woznicki et al. (2019,
False Alarm 0.22) and Wing et al. (2017, False Alarm 0.34-0.37). Critical Success Index (CSI) scores
penalize for over-prediction; our median value of 0.53 approximates previously published regional (e.g.,
Sangwan and Merwade, 2015, CSI values ranging from 0.44-0.89) and continental flood-extent
approaches (e.g., Sampson et al., 2015, CSI values from 0.43-0.67; Wing et al., 2017; CSI values between
0.50 and 0.55 reported). Median Precision (0.74) and F1 (0.70) values approximate those in the literature
as well (e.g., Woznicki et al., 2017 reported values of 0.78 for both). Mean Absolute Error of 0.08
reported here indicates an approximate 8 % difference between our GFPlain90 model and that of
Woznicki et al. (2017) at the 1 km cell resolution.

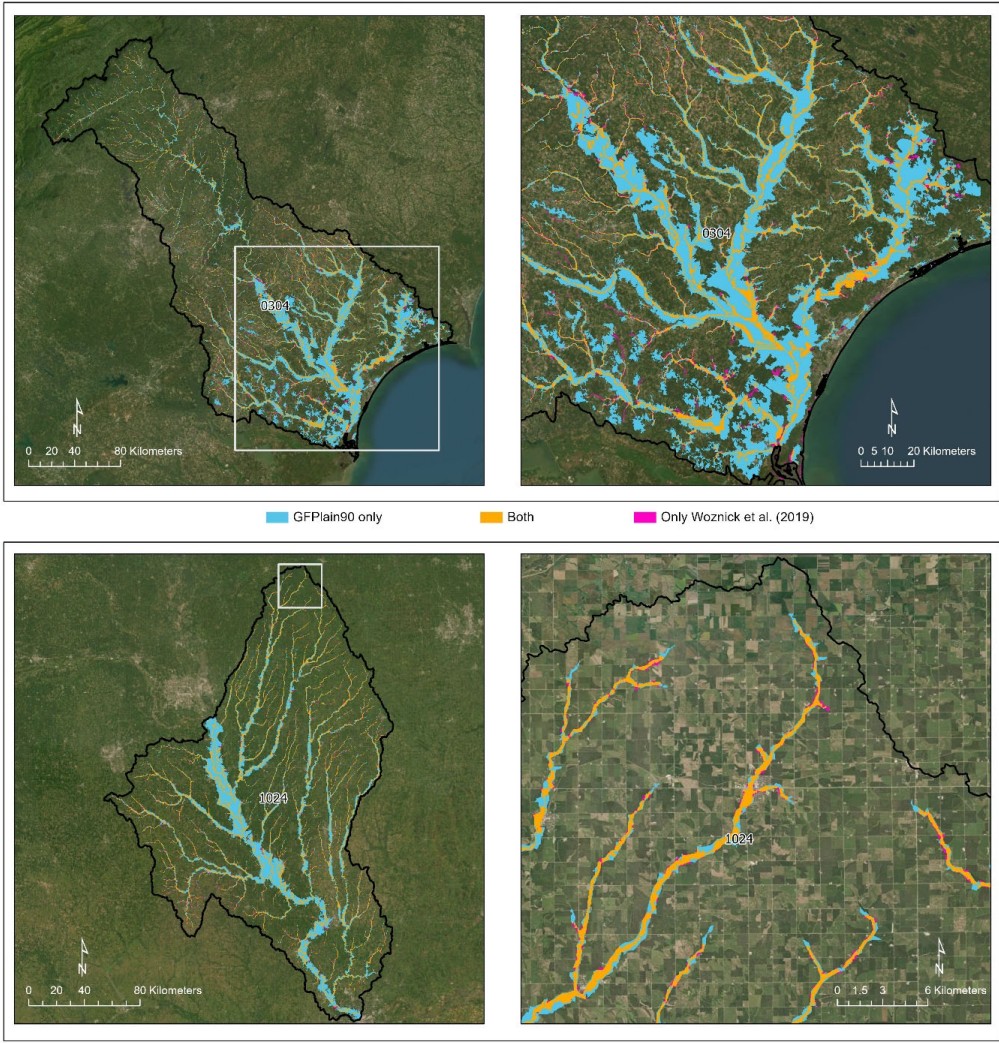

**Figure 3.** The robust performance of GFPlain90 relative to the benchmark Woznicki et al. (2019) floodplain data is

evident in the two rows, with the top panels (HUC_0304) a coastal watershed spanning North and South Carolina,

USA, and the bottom two panels different spatial extents of a midwestern USA watershed (HUC_1024). The

mainstem of the river network appeared wider in the GFPlain90 data in both examples, especially in the lower

reaches, though the complete network was well represented (i.e., floodplains were identified to the furthest extent of

the stream network's headwaters). Satellite imagery are sourced from ESRI (2022).



**Table 2.** Floodplain performance assessment of the GFPlain90-derived floodplain and the benchmark floodplain
from Woznicki et al. (2019). The first six equations directly assess the spatial concordance and overlap between the
two datasets, whereas Mean Absolute Error (Eq. 7) and Aggregate Error Bias (Eq. 8) are coarser fractional analyses
(i.e., the fraction of a 1 km$^2$ cell predicted correctly) as measured along the riverine network.

| Hydrologic Unit Code (HUC) ID | Hit Rate | Precision | False Alarm | CSI | F1 | Error Bias | Mean Absolute Error | Aggregate Error Bias |
|---|---|---|---|---|---|---|---|---|
| | (Eq. 1) | (Eq. 2) | (Eq. 3) | (Eq. 4) | (Eq. 5) | (Eq. 6) | (Eq. 7) | (Eq. 8) |
| HUC_0101 | 0.76 | 0.84 | 0.16 | 0.66 | 0.80 | 0.62 | 0.06 | -0.01 |
| HUC_0103 | 0.92 | 0.77 | 0.23 | 0.72 | 0.84 | 3.25 | 0.05 | 0.03 |
| HUC_0106 | 0.78 | 0.74 | 0.26 | 0.62 | 0.76 | 1.24 | 0.10 | -0.03 |
| HUC_0203 | 0.47 | 0.58 | 0.42 | 0.35 | 0.52 | 0.66 | 0.25 | -0.18 |
| HUC_0208 | 0.64 | 0.73 | 0.27 | 0.52 | 0.68 | 0.67 | 0.13 | -0.08 |
| HUC_0304 | 0.63 | 0.81 | 0.19 | 0.55 | 0.71 | 0.41 | 0.06 | 0.00 |
| HUC_0313 | 0.62 | 0.72 | 0.28 | 0.50 | 0.67 | 0.62 | 0.09 | -0.01 |
| HUC_0501 | 0.77 | 0.85 | 0.15 | 0.68 | 0.81 | 0.59 | 0.04 | -0.02 |
| HUC_0706 | 0.86 | 0.79 | 0.21 | 0.69 | 0.82 | 1.62 | 0.04 | 0.02 |
| HUC_0804 | 0.75 | 0.83 | 0.17 | 0.65 | 0.79 | 0.64 | 0.08 | -0.02 |
| HUC_1003 | 0.85 | 0.42 | 0.58 | 0.39 | 0.56 | 7.70 | 0.11 | 0.09 |
| HUC_1015 | 0.81 | 0.74 | 0.26 | 0.63 | 0.78 | 1.54 | 0.06 | 0.02 |
| HUC_1016 | 0.89 | 0.36 | 0.64 | 0.35 | 0.52 | 14.79 | 0.18 | 0.17 |
| HUC_1024 | 0.90 | 0.88 | 0.12 | 0.80 | 0.89 | 1.19 | 0.03 | 0.00 |
| HUC_1029 | 0.84 | 0.87 | 0.13 | 0.75 | 0.85 | 0.82 | 0.04 | -0.01 |
| HUC_1304 | 0.66 | 0.74 | 0.26 | 0.53 | 0.70 | 0.67 | 0.07 | -0.01 |
| HUC_1601 | 0.92 | 0.55 | 0.45 | 0.52 | 0.69 | 9.47 | 0.10 | 0.08 |
| HUC_1708 | 0.60 | 0.71 | 0.29 | 0.48 | 0.65 | 0.60 | 0.08 | -0.03 |
| HUC_1711 | 0.70 | 0.50 | 0.50 | 0.41 | 0.58 | 2.25 | 0.10 | -0.02 |
| HUC_1805 | 0.59 | 0.59 | 0.41 | 0.41 | 0.59 | 1.00 | 0.14 | -0.05 |
| HUC_1808 | 0.98 | 0.44 | 0.56 | 0.44 | 0.61 | 82.97 | 0.24 | 0.23 |
| Median | 0.77 | 0.74 | 0.26 | 0.53 | 0.70 | 1.00 | 0.08 | -0.01 |
| Mean | 0.76 | 0.69 | 0.31 | 0.56 | 0.71 | 6.35 | 0.10 | 0.01 |



**3.2**     **Wetland data performance**

**3.2.1**     **Global Wetland dataset**



The novel ensemble Global Wetlands approach improved upon the previously published Tootchi et al.
(2019) research product, the CW-WTD (Table 3) when contrasted with CONUS data. A median Hit Rate
value of 0.24 indicates that both the inclusive Global Wetlands and CW-WTD captured ~one-quarter of
the high-resolution, 30-m pixel size NLCD wetlands and open waters in the validation dataset. However,
across the 21 validation watersheds the Global Wetlands dataset developed here correctly identified more
wetlands than the CW-WTD alone, as indicated by an 8% mean increase in Precision, 43 % increase in
Critical Success Index, 38 % increase in F1, a -8 % decrease in the False Alarm ratio, and a 21 %
decrease in Error Bias. At coarser, 1 km$^2$ scales, there was a slight decrease in the Mean Absolute Error
associated with the Global Wetlands, and no difference in Aggregate Error Bias between the data
products.

**3.2.2    Global Non-Floodplain Wetland (Global NFW) dataset**

Non-floodplain wetland identification using the Global Wetlands data (i.e., Global NFWs) similarly
improved upon the CW-WTD product (Fig. 4). For instance, though the Hit Rate values were low (e.g.,
median values ≤ 0.10), underscoring both the difficulty in mapping non-floodplain wetlands and the
challenge of assessing performance using high-resolution data, Global NFW analyses correctly identified
50 % more non-floodplain wetlands than the CW-WTD (Table 4, Tootchi et al., 2019). Improvements
when focusing on non-floodplain wetlands were found in every category with the Global NFWs dataset,
demonstrating increased non-floodplain wetland accuracy versus the original CW-WTD across the
median metric values for Precision, Critical Success Index, F1, False Alarms, and Error Bias (e.g., 33 %
increase in Precision, 20 % increase in Critical Success Index, 10 % increase in F1 scores, and a 12 %





**Table 3.** Spatial performance assessment of both the Global Wetland (abbreviated here as GW) and CW-WTD
(abbreviated here as WTD, Tootchi et al., 2019) datasets when contrasted with the benchmark NLCD wetlands
(Dewitz, 2019). The first six equations directly assess the spatial concordance and overlap between each spatial
dataset and the benchmark (e.g., CW-WTD contrasted with the NLCD), whereas Mean Absolute Error (MAE, Eq.
7) and Aggregate Error Bias (AEB, Eq. 8) are coarser fractional analyses measured throughout each watershed (e.g.,
the proportional abundance NLCD within each 1 km$^2$ cell is contrasted with the proportional abundance of Global
Wetlands predicted correctly within that cell).

| Hydrologic Unit Code (HUC) ID | Hit Rate (Eq. 1) | | Precision (Eq. 2) | | False Alarm (Eq. 3) | | Critical Success (Eq. 4) | |
|---|---|---|---|---|---|---|---|---|
| | WTD | GW | WTD | GW | WTD | GW | WTD | GW |
| **HUC_0101** | 0.31 | 0.32 | 0.51 | 0.53 | 0.49 | 0.47 | 0.24 | 0.25 |
| **HUC_0103** | 0.26 | 0.28 | 0.42 | 0.45 | 0.58 | 0.55 | 0.19 | 0.21 |
| **HUC_0106** | 0.25 | 0.27 | 0.41 | 0.44 | 0.59 | 0.56 | 0.18 | 0.20 |
| **HUC_0203** | 0.12 | 0.12 | 0.51 | 0.53 | 0.49 | 0.47 | 0.11 | 0.11 |
| **HUC_0208** | 0.31 | 0.33 | 0.56 | 0.65 | 0.44 | 0.35 | 0.25 | 0.28 |
| **HUC_0304** | 0.42 | 0.43 | 0.65 | 0.69 | 0.35 | 0.31 | 0.35 | 0.36 |
| **HUC_0313** | 0.39 | 0.41 | 0.58 | 0.64 | 0.42 | 0.36 | 0.30 | 0.33 |
| **HUC_0501** | 0.15 | 0.17 | 0.57 | 0.64 | 0.43 | 0.36 | 0.14 | 0.15 |
| **HUC_0706** | 0.24 | 0.25 | 0.86 | 0.92 | 0.14 | 0.08 | 0.23 | 0.24 |
| **HUC_0804** | 0.45 | 0.46 | 0.70 | 0.75 | 0.30 | 0.25 | 0.38 | 0.40 |
| **HUC_1003** | 0.14 | 0.16 | 0.32 | 0.41 | 0.68 | 0.59 | 0.11 | 0.13 |
| **HUC_1015** | 0.25 | 0.40 | 0.17 | 0.42 | 0.83 | 0.58 | 0.12 | 0.26 |
| **HUC_1016** | 0.13 | 0.16 | 0.54 | 0.70 | 0.46 | 0.30 | 0.12 | 0.15 |
| **HUC_1024** | 0.10 | 0.10 | 0.67 | 0.75 | 0.33 | 0.25 | 0.09 | 0.10 |
| **HUC_1029** | 0.10 | 0.13 | 0.51 | 0.72 | 0.49 | 0.28 | 0.09 | 0.13 |
| **HUC_1304** | 0.02 | 0.02 | 0.44 | 0.52 | 0.56 | 0.48 | 0.02 | 0.02 |
| **HUC_1601** | 0.29 | 0.33 | 0.34 | 0.45 | 0.66 | 0.55 | 0.19 | 0.24 |
| **HUC_1708** | 0.24 | 0.24 | 0.48 | 0.49 | 0.52 | 0.51 | 0.19 | 0.20 |
| **HUC_1711** | 0.09 | 0.10 | 0.46 | 0.51 | 0.54 | 0.49 | 0.08 | 0.09 |
| **HUC_1805** | 0.14 | 0.15 | 0.62 | 0.64 | 0.38 | 0.36 | 0.13 | 0.13 |
| **HUC_1808** | 0.12 | 0.13 | 0.51 | 0.55 | 0.49 | 0.45 | 0.11 | 0.11 |
| **Median** | 0.24 | 0.24 | 0.51 | 0.55 | 0.49 | 0.45 | 0.14 | 0.20 |
| **Difference** | | 0.00 | | 0.04 | | -0.04 | | 0.06 |
| **Change (%)** | | 0.0 | | 7.8 | | -8.2 | | 42.9 |




**Table 3.** (Continued)

| Hydrologic Unit Code (HUC) ID | F1 (Eq. 5) | | Error Bias (Eq. 6) | | MAE (Eq. 7) | | AEB (Eq. 8) | |
|---|---|---|---|---|---|---|---|---|
| | WTD | GW | WTD | GW | WTD | GW | WTD | GW |
| HUC_0101 | 0.38 | 0.40 | 0.43 | 0.40 | 0.18 | 0.17 | 0.09 | 0.09 |
| HUC_0103 | 0.32 | 0.34 | 0.50 | 0.47 | 0.16 | 0.15 | 0.06 | 0.07 |
| HUC_0106 | 0.31 | 0.33 | 0.49 | 0.46 | 0.20 | 0.19 | 0.08 | 0.08 |
| HUC_0203 | 0.19 | 0.20 | 0.13 | 0.13 | 0.36 | 0.36 | 0.28 | 0.28 |
| HUC_0208 | 0.40 | 0.44 | 0.35 | 0.27 | 0.17 | 0.17 | 0.09 | 0.10 |
| HUC_0304 | 0.51 | 0.53 | 0.39 | 0.34 | 0.21 | 0.21 | 0.12 | 0.13 |
| HUC_0313 | 0.47 | 0.50 | 0.48 | 0.38 | 0.16 | 0.16 | 0.07 | 0.09 |
| HUC_0501 | 0.24 | 0.26 | 0.14 | 0.11 | 0.10 | 0.10 | 0.09 | 0.09 |
| HUC_0706 | 0.37 | 0.39 | 0.05 | 0.03 | 0.12 | 0.12 | 0.11 | 0.12 |
| HUC_0804 | 0.55 | 0.57 | 0.36 | 0.29 | 0.20 | 0.20 | 0.12 | 0.14 |
| HUC_1003 | 0.19 | 0.23 | 0.34 | 0.27 | 0.04 | 0.04 | 0.02 | 0.02 |
| HUC_1015 | 0.21 | 0.41 | 1.59 | 0.91 | 0.05 | 0.04 | -0.01 | 0.00 |
| HUC_1016 | 0.21 | 0.26 | 0.13 | 0.08 | 0.26 | 0.26 | 0.23 | 0.24 |
| HUC_1024 | 0.17 | 0.18 | 0.05 | 0.04 | 0.14 | 0.14 | 0.13 | 0.14 |
| HUC_1029 | 0.17 | 0.22 | 0.11 | 0.06 | 0.11 | 0.12 | 0.10 | 0.11 |
| HUC_1304 | 0.04 | 0.04 | 0.02 | 0.02 | 0.08 | 0.08 | 0.08 | 0.08 |
| HUC_1601 | 0.31 | 0.38 | 0.80 | 0.59 | 0.05 | 0.05 | 0.01 | 0.01 |
| HUC_1708 | 0.32 | 0.33 | 0.34 | 0.33 | 0.15 | 0.15 | 0.08 | 0.08 |
| HUC_1711 | 0.15 | 0.17 | 0.12 | 0.11 | 0.17 | 0.17 | 0.14 | 0.15 |
| HUC_1805 | 0.23 | 0.24 | 0.10 | 0.10 | 0.25 | 0.26 | 0.21 | 0.21 |
| HUC_1808 | 0.20 | 0.20 | 0.13 | 0.12 | 0.09 | 0.09 | 0.07 | 0.07 |
| Median | 0.24 | 0.33 | 0.34 | 0.27 | 0.16 | 0.15 | 0.09 | 0.09 |
| Difference | | 0.09 | | 0.07 | | -0.01 | | 0.00 |
| Change (%) | | 37.5 | | -20.6 | | -6.3 | | 0.0 |



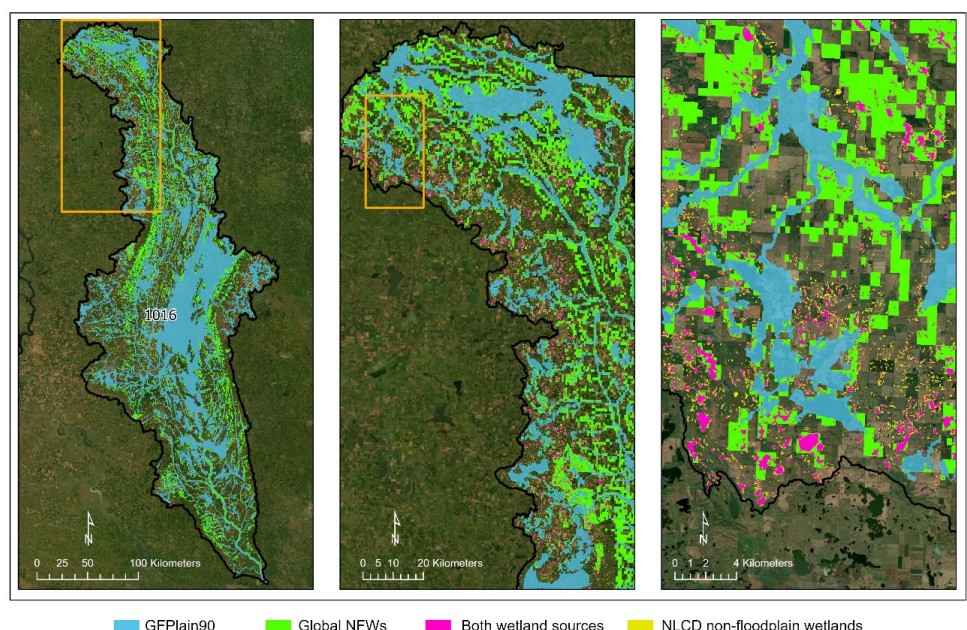


**Figure 4.** Demonstration of the relative accuracy of the Global NFWs in identifying non-floodplain wetlands using a

Prairie Pothole Region watershed (HUC_1016, see Fig. 2) replete with abundant non-floodplain wetlands. Correctly

identified wetlands occur in both wetland sources (magenta color). Omission errors (NLCD non-floodplain

wetlands, smaller systems in yellow) and commission errors (Global NFWs, green) are evident as a result of the

higher resolution of the NLCD validation dataset. Satellite imagery are sourced from ESRI (2022).





**Table 4.** Non-floodplain wetland performance metrics contrasting both the Global NFWs (abbreviated here as
GNFW) and CW-WTD (abbreviated here as WTD, Tootchi et al., 2019) non-floodplain wetland spatial data with the
benchmark NLCD wetlands (Dewitz, 2019). Descriptions of the metrics are the same as in Table 3, though the focus
here is on wetlands outside the GFPlain90-derived floodplain.

| Hydrologic Unit Code (HUC) ID | Hit Rate (Eq. 1) | | Precision (Eq. 2) | | False Alarm (Eq. 3) | | Critical Success Index (Eq. 4) | |
|---|---|---|---|---|---|---|---|---|
| | WTD | GNFW | WTD | GNFW | WTD | GNFW | WTD | GNFW |
| **HUC_0101** | 0.24 | 0.25 | 0.43 | 0.45 | 0.57 | 0.55 | 0.18 | 0.19 |
| **HUC_0103** | 0.17 | 0.18 | 0.30 | 0.32 | 0.70 | 0.68 | 0.12 | 0.13 |
| **HUC_0106** | 0.15 | 0.18 | 0.14 | 0.17 | 0.86 | 0.83 | 0.08 | 0.10 |
| **HUC_0203** | 0.12 | 0.13 | 0.20 | 0.23 | 0.80 | 0.77 | 0.08 | 0.09 |
| **HUC_0208** | 0.14 | 0.16 | 0.34 | 0.41 | 0.66 | 0.59 | 0.11 | 0.13 |
| **HUC_0304** | 0.26 | 0.28 | 0.45 | 0.49 | 0.55 | 0.51 | 0.20 | 0.21 |
| **HUC_0313** | 0.21 | 0.23 | 0.35 | 0.40 | 0.65 | 0.60 | 0.15 | 0.17 |
| **HUC_0501** | 0.05 | 0.07 | 0.32 | 0.41 | 0.68 | 0.59 | 0.05 | 0.06 |
| **HUC_0706** | 0.05 | 0.06 | 0.63 | 0.72 | 0.37 | 0.28 | 0.05 | 0.05 |
| **HUC_0804** | 0.30 | 0.31 | 0.51 | 0.55 | 0.49 | 0.45 | 0.23 | 0.25 |
| **HUC_1003** | 0.04 | 0.07 | 0.13 | 0.21 | 0.87 | 0.79 | 0.03 | 0.05 |
| **HUC_1015** | 0.07 | 0.25 | 0.05 | 0.28 | 0.95 | 0.72 | 0.03 | 0.15 |
| **HUC_1016** | 0.07 | 0.11 | 0.31 | 0.53 | 0.69 | 0.47 | 0.06 | 0.10 |
| **HUC_1024** | 0.02 | 0.04 | 0.18 | 0.41 | 0.82 | 0.59 | 0.02 | 0.04 |
| **HUC_1029** | 0.03 | 0.06 | 0.25 | 0.58 | 0.75 | 0.42 | 0.03 | 0.06 |
| **HUC_1304** | 0.00 | 0.00 | 0.26 | 0.33 | 0.74 | 0.67 | 0.00 | 0.00 |
| **HUC_1601** | 0.05 | 0.09 | 0.07 | 0.16 | 0.93 | 0.84 | 0.03 | 0.06 |
| **HUC_1708** | 0.06 | 0.06 | 0.33 | 0.35 | 0.67 | 0.65 | 0.05 | 0.05 |
| **HUC_1711** | 0.04 | 0.05 | 0.22 | 0.27 | 0.78 | 0.73 | 0.04 | 0.05 |
| **HUC_1805** | 0.06 | 0.07 | 0.27 | 0.30 | 0.73 | 0.70 | 0.05 | 0.06 |
| **HUC_1808** | 0.05 | 0.06 | 0.25 | 0.36 | 0.75 | 0.64 | 0.04 | 0.06 |
| **Median** | 0.06 | 0.09 | 0.27 | 0.36 | 0.73 | 0.64 | 0.05 | 0.06 |
| **Difference** | | 0.03 | | 0.09 | | -0.09 | | 0.01 |
| **Change (%)** | | 50.0 | | 33.3 | | -12.3 | | 20.0 |





**Table 4.** (Continued)

| Hydrologic Unit Code (HUC) ID | F1 (Eq. 5) | | Error Bias (Eq. 6) | | Mean Absolute Error (Eq. 7) | | Aggregate Error Bias (Eq. 8) | |
|---|---|---|---|---|---|---|---|---|
| | WTD | GNFW | WTD | GNFW | WTD | GNFW | WTD | GNFW |
| HUC_0101 | 0.31 | 0.32 | 0.41 | 0.39 | 0.16 | 0.16 | 0.08 | 0.09 |
| HUC_0103 | 0.21 | 0.23 | 0.48 | 0.46 | 0.14 | 0.14 | 0.06 | 0.06 |
| HUC_0106 | 0.14 | 0.18 | 1.11 | 1.03 | 0.11 | 0.11 | -0.01 | 0.00 |
| HUC_0203 | 0.15 | 0.17 | 0.53 | 0.51 | 0.09 | 0.09 | 0.03 | 0.03 |
| HUC_0208 | 0.20 | 0.23 | 0.32 | 0.28 | 0.10 | 0.10 | 0.07 | 0.07 |
| HUC_0304 | 0.33 | 0.35 | 0.44 | 0.40 | 0.17 | 0.17 | 0.08 | 0.09 |
| HUC_0313 | 0.27 | 0.29 | 0.51 | 0.45 | 0.12 | 0.12 | 0.05 | 0.06 |
| HUC_0501 | 0.09 | 0.11 | 0.12 | 0.10 | 0.09 | 0.09 | 0.07 | 0.08 |
| HUC_0706 | 0.09 | 0.10 | 0.03 | 0.02 | 0.09 | 0.09 | 0.09 | 0.09 |
| HUC_0804 | 0.38 | 0.40 | 0.43 | 0.37 | 0.13 | 0.13 | 0.07 | 0.07 |
| HUC_1003 | 0.07 | 0.10 | 0.32 | 0.26 | 0.02 | 0.03 | 0.01 | 0.01 |
| HUC_1015 | 0.06 | 0.26 | 1.46 | 0.85 | 0.03 | 0.02 | -0.01 | 0.00 |
| HUC_1016 | 0.11 | 0.19 | 0.17 | 0.11 | 0.15 | 0.15 | 0.12 | 0.13 |
| HUC_1024 | 0.03 | 0.07 | 0.09 | 0.06 | 0.05 | 0.05 | 0.04 | 0.05 |
| HUC_1029 | 0.05 | 0.11 | 0.09 | 0.05 | 0.08 | 0.08 | 0.07 | 0.07 |
| HUC_1304 | 0.00 | 0.01 | 0.01 | 0.01 | 0.06 | 0.06 | 0.05 | 0.06 |
| HUC_1601 | 0.05 | 0.11 | 0.68 | 0.50 | 0.02 | 0.02 | 0.00 | 0.01 |
| HUC_1708 | 0.10 | 0.10 | 0.12 | 0.12 | 0.11 | 0.11 | 0.09 | 0.10 |
| HUC_1711 | 0.07 | 0.09 | 0.16 | 0.15 | 0.09 | 0.09 | 0.07 | 0.07 |
| HUC_1805 | 0.10 | 0.11 | 0.18 | 0.17 | 0.10 | 0.10 | 0.07 | 0.07 |
| HUC_1808 | 0.08 | 0.11 | 0.15 | 0.12 | 0.02 | 0.02 | 0.01 | 0.01 |
| **Median** | 0.10 | 0.11 | 0.32 | 0.26 | 0.09 | 0.09 | 0.07 | 0.07 |
| **Difference** | | 0.01 | | -0.06 | | 0.00 | | 0.00 |
| **Change (%)** | | 10.0 | | -18.8 | | 0.0 | | 0.0 |


decrease in False Alarms and a 19 % decrease in Error Bias). There was no difference between the
datasets with median values for Mean Absolute Error (median values for both = 0.09) or Aggregate Error
Bias (median values for both = 0.07). Thus, at the 1 km² cell size, there was <10 % difference between
both the CW-WTD and the Global NFWs and the benchmark NLCD non-floodplain wetlands and open
waters (with the difference mostly stemming from an increase in identified wetlands with both CW-WTD
and Global NFWs, as indicated with the positive Aggregate Error Bias values).



**3.3     Global extent analyses and synthesis**

**3.3.1     Floodplains**

Floodplains were estimated to cover 26.6 million $km^2$ (Table 5), or 19.7 % of the global landmass.
Approximately 23-24 % of the African and Australasian land masses were categorized as occurring
within a floodplain, the greatest percentage of global areas so categorized. Conversely, the Arctic
(northern Canada and Alaska) and Greenland (excluding the ice sheet) had the least land mass categorized
as floodplain (13-14 %). In comparison, Nardi et al. (2019) calculated a global floodplain extent of
13,394,139 $km^2$, using a 250-m pixel size, a 1000 $km^2$ minimum contributing area, and bounding their
study between 60° N and 60° S latitudes. Our analyses using the same latitudinal bounds but with a higher
resolution dataset (90 m) and a 20 $km^2$ minimum contributing area identified 24,185,775 $km^2$, an 81 %
areal increase (Fig. B1).

**Table 5.** Calculated floodplain area for each HydroBASINS at the global scale. Our analyses found 19.7 % of the
landmass occurs within a floodplain.

| HydroBASINS Region | Floodplain ($km^2$) | Floodplain Percent of Landmass |
|---|---|---|
| Africa | 6,990,859 | 23.3 % |
| Arctic (northern Canada & Alaska) | 894,594 | 14.2 % |
| Asia | 4,283,991 | 20.6 % |
| Australasia | 2,649,395 | 23.8 % |
| Europe and Middle East | 3,415,308 | 19.1 % |
| Greenland (excl. ice sheet) | 270,813 | 12.6 % |
| North & Central America (excl. Alaska) | 2,713,346 | 17.0 % |
| Siberian Russia | 2,051,305 | 15.8 % |
| South America | 3,368,778 | 18.9 % |
| Total | 26,638,389 | 19.7 % |



**3.3.2     Wetlands**



Global Wetland extent covered 30.5 million km$^2$ (Table 6). With a focus on smaller systems compared to
those presented by Tootchi et al. (2019), our Global Wetland dataset identified 11 % more potential
global wetlands (3 million km$^2$ additional wetlands).

Australasia had the greatest proportional wetland abundance (see also Zhu et al., 2022), with wetlands
covering 38 % of the landmass (driven, in part, by island abundance and fringing estuarine wetlands [Fan
et al., 2013]). Greenland (3 %) and Africa (12 %) had the least wetlands identified on the land mass.

**Table 6.** Estimated Global Wetlands areal extent for each of the nine regional HydroBASINS (Lehner and Grill,
2013). As described in the text, Global Wetlands extent incorporates the CW-WTD (Tootchi et al., 2019), CCI
(Herold et al., 2015), and GSW (Pekel et al., 2016); lakes of ≥ 10 ha have been removed (Messager et al., 2016).

| HydroBASINS Region | Wetlands (km$^2$) | Wetland Percent of Landmass |
|---|---|---|
| Africa | 3,524,917 | 11.8 % |
| Arctic (northern Canada & Alaska) | 1,807,830 | 28.6 % |
| Asia | 5,543,333 | 26.6 % |
| Australasia | 4,283,996 | 38.4 % |
| Europe and Middle East | 2,465,074 | 13.8 % |
| Greenland (excl. ice sheet) | 60,761 | 2.8 % |
| North & Central America (excl. Alaska) | 4,107,333 | 25.8 % |
| Siberian Russia | 3,578,868 | 27.6 % |
| South America | 5,140,139 | 28.8 % |
| Total | 30,512,251 | 22.6 % |



**3.3.3    Non-floodplain wetlands (Global NFW)**

Approximately 16.0 million km$^2$ of potential non-floodplain wetlands were identified globally (Global
NFWs, Fig. 5), meaning that 11.9 % of the global landmass is estimated to be covered by non-floodplain
wetlands (Table 7). This represents ~53 % of the total global wetlands found in the dataset used in this
analysis (see Methods: Wetland Data, above). The global distribution of non-floodplain wetlands is
widespread, though they were found to comprise a higher proportion of wetlands within more northern

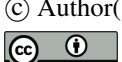

HydroBASINS watersheds (i.e., higher abundances in formerly glaciated basins), as demonstrated in Fig.
6. The Arctic portion of northern Canada and Alaska (21.7 %), and Siberian Russia (17.4 %), typically
underlain by permafrost and frequently inundated or saturated due to poor drainage evolution
(Kremenetski et al., 2003; Robarts et al., 2013; Olefeldt et al., 2021), had the greatest percent non-

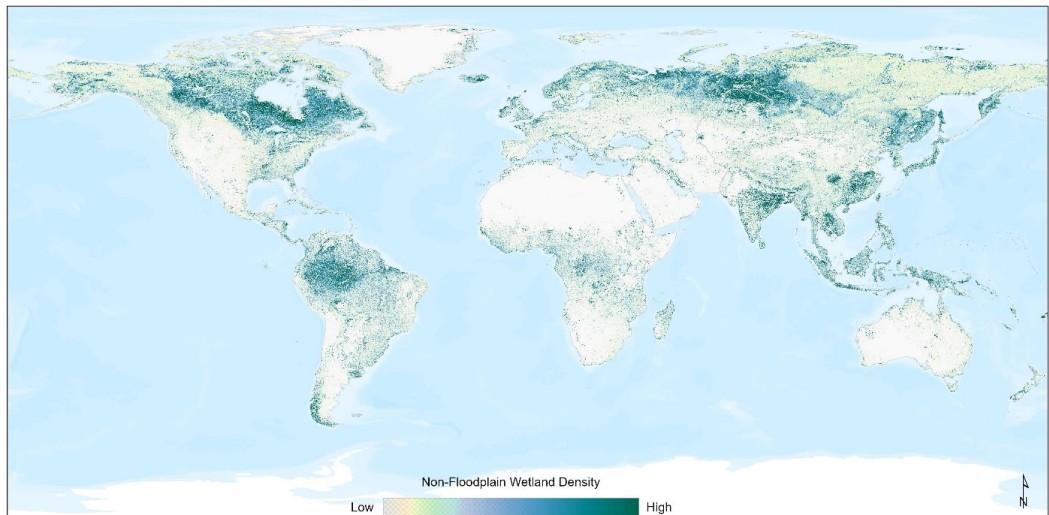


**Figure 5.** Non-floodplain wetlands, Global NFWs, are found worldwide, with a greater abundance in formerly
glaciated landscapes of northern climates (e.g., northern North America and Siberian Russia) as well as within the
Amazon basin (South America). This density map was created using the Focal Statistics tool in ArcGIS Pro 2.9.1.
The basemap layer is the ESRI World Terrain Base (2022).





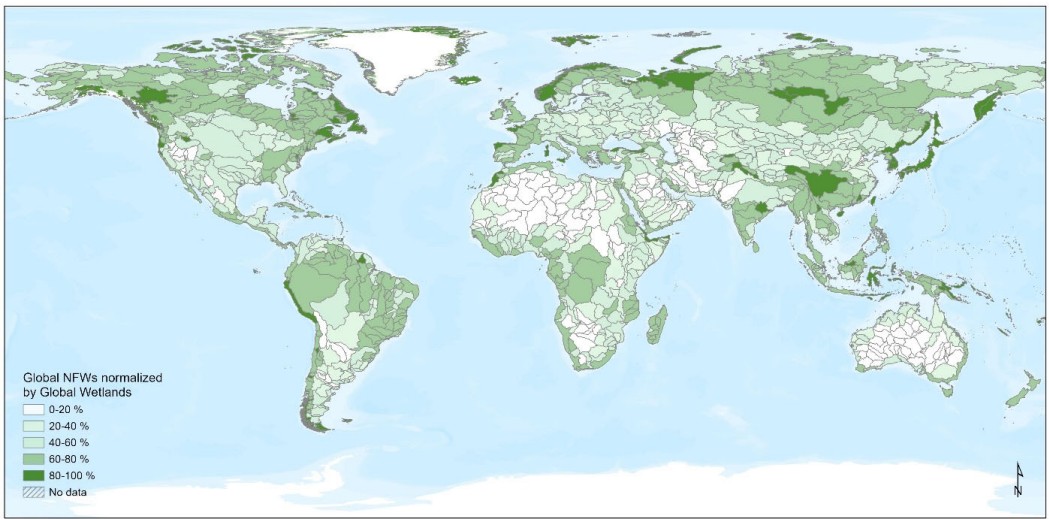


**Figure 6.** The proportion of non-floodplain wetlands, Global NFWs, within a given HydroBASINS watershed

(Lehrner and Grill, 2013), ranging up to 100 %, varied globally. The impacts or effects of non-floodplain wetlands

on biological, biogeochemical, and hydrological functions will vary based on their relative abundance, location

within the watershed, and hydrologic characteristics (Lane et al., 2018). The basemap layer is the ESRI World

Terrain Base (2022).

567

568

floodplain wetlands. Africa (5.4 %) and Greenland (1.0 %, excluding ice sheets) had the least abundance

of non-floodplain wetlands. A four-direction region-group (contagion) analysis conducted to identify

adjacent pixels considered as contiguous units or non-floodplain wetland systems identified 32.8 million

individual non-floodplain wetlands. Non-floodplain wetlands are typically small aquatic systems (see

Table 7): the median size differed across the HydroBASINS regions from 0.018 km$^2$ (1.8 ha) to 0.138

km$^2$ (13.8 ha) with a global median of 0.039 km$^2$ (3.87 ha).



**Table 7.** Global NFW data further described by HydroBASIN region.

| HydroBASINS Region | Global NFW Extent (km²) | Count of Global NFWs (#) | Global NFW Percent of Landmass | Global NFW Median Area (km²) |
|---|---|---|---|---|
| Africa | 1,611,225 | 2,698,465 | 5.4 % | 0.138 |
| Arctic (northern Canada & Alaska) | 1,371,937 | 5,956,081 | 21.7 % | 0.018 |
| Asia | 2,924,900 | 4,564,172 | 14.0 % | 0.049 |
| Australasia | 850,402 | 1,448,315 | 7.6 % | 0.054 |
| Europe and Middle East | 1,475,355 | 3,740,961 | 8.3 % | 0.054 |
| Greenland (excl. ice sheet) | 21,747 | 180,726 | 1.0 % | 0.018 |
| North & Central America (excl. Alaska) | 2,608,158 | 5,740,066 | 16.4 % | 0.025 |
| Siberian Russia | 2,255,689 | 4,864,577 | 17.4 % | 0.063 |
| South America | 2,891,604 | 3,572,294 | 16.2 % | 0.096 |
| **Total** | 16,011,018 | 32,765,657 | 11.9 % | 0.039 |

## 4    Discussion

We report here for the first time the global abundance of non-floodplain wetlands, a functionally

important and imperiled resource (Creed et al., 2017). Our estimate of 16.0 million km² suggests that

approximately 53 % of the global population of wetlands are likely non-floodplain wetland systems.

These aquatic systems are small, with a range from 0.018-0.138 km² (1.8-13.8 ha) across the globe and a

global median size of 0.039 km² (3.87 ha, see Table 7).

The global abundance of non-floodplain wetlands is a reasonable first approximation of the total non-

floodplain wetland extent. For instance, non-floodplain wetland estimates in the CONUS were conducted

by Lane et al. (2022) using high-resolution aerial-sourced spatial data layers developed by the National

Wetlands Inventory (U.S. Fish and Wildlife Service, various dates). Lane et al. (2022) reported

approximately 23% of the area of freshwater wetlands to be non-floodplain wetland systems. Yet the

CONUS has lost nearly half of its wetlands since the European colonization (Dahl, 1990), with smaller

and shallower non-floodplain wetlands likely being disproportionately lost (Van Meter and Basu, 2015;

Serran et al., 2017).



Tootchi et al. (2019) – our base input geospatial data layer – calculated that the global wetland extent
identified from incorporating both regularly flooded wetland systems (surface-water and precipitation-
sourced) and groundwater-driven wetland systems (e.g., Fan et al., 2013; Hu et al., 2017b) resulted in
approximately 27.5 million km$^2$ of wetlands, a value towards the higher-end of previously published
geospatial wetland datasets (Hu et al., 2017a). In their synthesis, Tootchi et al. (2019) explained their
values as particularly influenced by groundwater-driven wetlands, especially those in the tropics (10$^o$ N-
10$^o$ S latitudes, Zhu et al., 2022), following recent studies acknowledging the under-estimation of those
wetland systems (e.g. Wania et al., 2013; Gumbricht et al., 2017).

It follows that incorporating additional higher-resolution satellite inundation data (Pekel et al., 2016) as
well as groundwater-driven wetland systems data (e.g., Fan et al., 2013; Tootchi et al., 2019), as
conducted in this study, would similarly maintain the trend towards the higher end in global estimates as
found by Hu et al. (2017a) and Toochi et al. (2019). This is meted out in the simple contrast between the
proportional abundance of non-floodplain wetland systems identified here against the 30 m NLCD data
product described above (Dewitz, 2019) across the 21 CONUS watersheds in this study. The calculated
median watershed abundance of non-floodplain wetlands in both the Global NFWs (9.4 %) and the
Tootchi et al. (2019) CW-WTD (9.1 %) datasets from our validation watersheds are nearly 5-fold the
abundance of the benchmark data from the NLCD (Table 7). However, this is contrasted with a 7-fold
*under-representation* of non-floodplain wetlands as derived from the satellite based GSW data (Table 8,
Pekel et al., 2016).

It is apparent that the GSW alone is insufficient to map non-floodplain wetlands (this study, Vanderhoof
and Lane, 2019). Though useful as a satellite-based input data layer, the GSW by itself appears
inadequate for identifying non-floodplain wetlands because it relies on surface-water inundation and
ignores saturated wetland systems and those driven by groundwater discharge and upwelling (Winter et
al., 1998). Fan et al. (2013) found that groundwater drivers of aquatic system state were important and





**Table 8.** A comparison of the non-floodplain wetland distribution within the 21 HUCs contrasting across NLCD
(the benchmark data layer, Dewitz, 2019), Global NFW (this study), CW-WTD (Tootchi et al., 2019), and GSW
(Pekel et al., 2016). The CW-WTD (at 500 m) and the Global NFW (coupling 500 m, 300 m, and 30 m data),
derived from the CW-WTD, identified 5-fold the abundance of non-floodplain wetlands whereas the GSW under-
estimated non-floodplain wetlands nearly 7-fold.

| HUC ID | Percent HUC as NLCD NFW | Percent HUC as Global NFW | Percent HUC as CW-WTD NFW | Percent HUC as GSW NFW |
|---|---|---|---|---|
| HUC_0101 | 10.4 % | 19.2 % | 18.9 % | 0.1 % |
| HUC_0103 | 8.1 % | 14.6 % | 14.3 % | 0.2 % |
| HUC_0106 | 8.2 % | 8.0 % | 7.5 % | 0.3 % |
| HUC_0203 | 4.9 % | 8.4 % | 8.3 % | 0.4 % |
| HUC_0208 | 4.7 % | 12.0 % | 11.5 % | 4.6 % |
| HUC_0304 | 12.2 % | 21.7 % | 20.8 % | 12.2 % |
| HUC_0313 | 8.3 % | 14.4 % | 13.5 % | 8.2 % |
| HUC_0501 | 1.5 % | 9.2 % | 8.9 % | 0.1 % |
| HUC_0706 | 0.7 % | 9.7 % | 9.5 % | 0.3 % |
| HUC_0804 | 9.7 % | 17.1 % | 16.2 % | 9.7 % |
| HUC_1003 | 0.7 % | 2.1 % | 1.9 % | 0.2 % |
| HUC_1015 | 1.8 % | 2.0 % | 1.3 % | 0.1 % |
| HUC_1016 | 3.6 % | 16.6 % | 15.5 % | 2.6 % |
| HUC_1024 | 0.5 % | 5.1 % | 4.9 % | 0.2 % |
| HUC_1029 | 0.9 % | 8.4 % | 7.7 % | 0.4 % |
| HUC_1304 | 0.0 % | 5.5 % | 5.5 % | 0.0 % |
| HUC_1601 | 0.7 % | 1.3 % | 1.0 % | 0.1 % |
| HUC_1708 | 2.0 % | 11.7 % | 11.6 % | 0.4 % |
| HUC_1711 | 1.8 % | 9.4 % | 9.1 % | 0.2 % |
| HUC_1805 | 2.2 % | 9.9 % | 9.7 % | 0.5 % |
| HUC_1808 | 0.3 % | 1.7 % | 1.6 % | 0.1 % |
| **Median** | 2.0 % | 9.4 % | 9.1 % | 0.3 % |


underrepresented in global datasets. Relying on surface water inundation captured during satellite
overflights depends not only on an unobstructed view of the waterbody (e.g., not obscured by trees) but
also fortuitous timing regarding inundation status. For example, in an analysis of non-floodplain wetlands
of the CONUS as derived by distance from an aquatic system, Lane and D'Amico (2016) reported that
just over 50 % of the non-floodplain wetlands were classified as seasonally or temporarily flooded –
meaning that cloud-free and unobscured overflights would only potentially identify these systems at
certain inundated times of the year. Additionally, Lane and D'Amico (2016) identified another 6 % of
CONUS non-floodplain wetlands as saturated (i.e., wetlands with saturated substrates but with surface
water seldom present). These wetlands would not be identified by the GSW (Pekel et al., 2016) resulting



in a further under-representation of the global resource. Similarly, Hamunyela et al. (2022), analyzing
~150,000 km$^2$ in southeastern Africa, found that the GSW underestimated surface water extent (i.e.,
omission errors) by nearly 65%. Vanderhoof and Lane (2019) found approximately 42% omission rates
when contrasting the GSW data to surface-water extent in non-floodplain wetlands ranging from 0.2-17.6
ha in area in the Midwestern US. While the GSW is an outstanding dataset that is continuing to be
managed and updated, the GSW and its derived product have limitations in their stand-alone utility in
global non-floodplain wetland analyses.

While solely using satellite-based surface-water data products omits groundwater-driven and saturated
wetlands and likely results in non-floodplain wetland underestimations, our Global Wetland data
incorporated the finer-resolution CCI (Herold et al., 2015) and GSW (Pekel et al., 2016) products into the
Tootchi et al. (2019) base map, substantially improving the identification of non-floodplain wetlands.
These improvements, as indicated by performance indices increasing from 10-50 % in the derived Global
Wetland data (see Table 4), support the inclusion of these higher-resolution satellite-based data (Herold et
al., 2015; Pekel et al., 2016) with groundwater datasets (Fan et al., 2013), especially when focused on
smaller and non-floodplain wetland systems. Similarly, at a coarser scale of 1 km, there was a difference
in Mean Absolute Error value of 0.09 (see Table 4) between the Global NFWs and the benchmark NLCD.
This ~9 % difference between the two datasets at a 1 km resolution (the former originating at 500 m and
the latter at 30 m) further suggest substantive potential utility in these global non-floodplain wetland data
for effective natural resource management and decision-making.




## 5    Implications

Non-floodplain wetlands remain vulnerable waters (Creed et al., 2017), despite the fact that the
hydrological, biogeochemical, and biological functions performed by non-floodplain wetlands are
increasingly noted in the literature (e.g., Leibowitz, 2003; Creed et al., 2017; Lane et al., 2018; Lane et
al., 2022), incorporated into eco-hydrological models by the scientific community (e.g., Fossey and
Rousseau, 2016; Golden et al., 2017; Golden et al., 2021; Leibowitz et al., In Review), and considered by
policy makers (e.g., Biggs et al., 2017; Drenkhan et al., 2022). Their global fate has important
implications for watershed-scale resilience to changing climatic conditions (Mckenna et al., 2017; Lane et
al., 2022) affecting the measured benefits humans receive from biogeochemical processing, stormwater
attenuation, and drought mitigation functions provided by non-floodplain wetlands.

Global attention to functions of non-floodplain wetlands has increased in the United States (Marton et al.,
2015; Rains et al., 2016; Cohen et al., 2016), Europe (Biggs et al., 2017; Nitzsche et al., 2017; Rodríguez-
Rodríguez et al., 2021), Asia (Kam, 2010; Van Meter et al., 2014), Australia (Adame et al., 2019), Africa
(Merken et al., 2015; Samways et al., 2020), South America (Rodrigues et al., 2012; Cunha et al., 2019)
and elsewhere (see extensive review in Chen et al., 2022). This includes analyses of non-floodplain
wetlands both as individual systems (e.g., assessing the functions of a single wetland or wetland complex;
Badiou et al., 2018) as well as agglomerated, watershed-scale functioning systems (e.g., answering
questions on the functional contributions of all non-floodplain wetlands at larger spatial extents; Golden
et al., 2016; Blanchette et al., 2022). Previous studies found that non-floodplain wetlands are
overwhelmingly important contributors to biogeochemical and hydrological functions affecting
downgradient (i.e., down-stream) water quality and streamflow (e.g., McLaughlin et al., 2014; Marton et
al., 2015; Cohen et al., 2016; Rains et al., 2016; Golden et al., 2019; Cheng et al., 2020). Hence, with the
development of this publicly available dataset, and subsequent improvements by others, it is hoped that





these important aquatic systems will be incorporated into resource management and decision-making
across the globe.

Recently, Lane et al. (2022) identified global-scale geospatial data of the spatial extent and spatial
configuration of vulnerable waters – non-floodplain wetlands and headwater stream systems (e.g.,
ephemeral, intermittent, and perennial low-order waters [Strahler, 1957]) – as a critical scientific gap.
Discounting their significance in watershed-scale hydrology and nutrient biogeochemistry analyses – as
well as their importance in biological processes (Schofield et al., 2018; Smith et al., 2019; Mushet et al.,
2019) – affects quantification of the myriad ecosystem services they provide (De Groot, 2006; Colvin et
al., 2019). For instance, Golden et al. (2021) provide a tangible example of the functional effects and
influence of non-floodplain wetlands once incorporated into watershed-scale hydrologic models (Fig. 7):
ignoring non-floodplain wetlands in the model resulted in projected critical flood-stage return intervals
(e.g., 50 yr and 100 yr floods) being reached within a given modeled time frame. Conversely,
incorporating non-floodplain wetlands and their storage capacities into a river basin model (e.g., Rajib et
al., 2020) demonstrated that non-floodplain wetlands significantly attenuate storm flows, for when non-
floodplain wetlands are "…integrated into the model, those simulated *flood stages are not reached*"
(Golden et al., 2021, p. 3, emphasis added). The hydrological functions and concomitantly the associated
biogeochemical functions (e.g., Marton et al., 2015) of non-floodplain wetlands demand an effective
accounting of their spatial extent and configuration, as demonstrated in this novel global dataset.

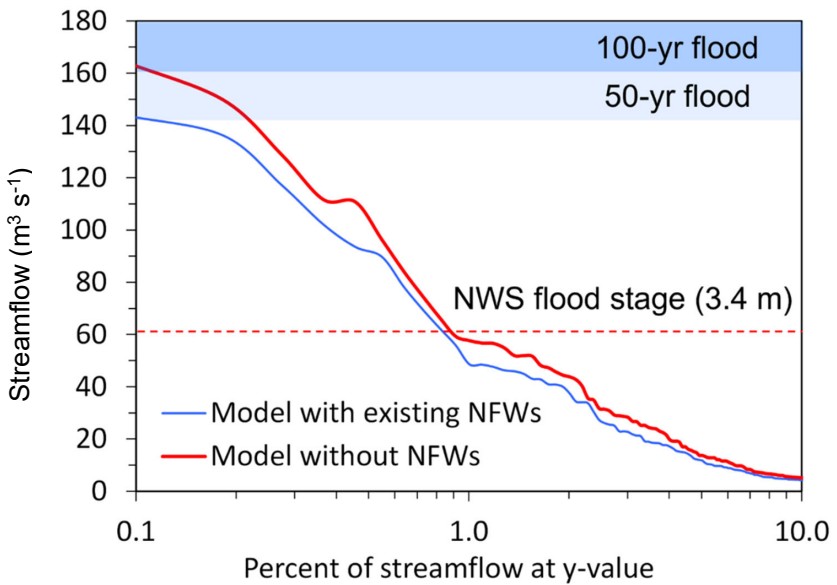

**Figure 7.** Non-floodplain wetlands attenuate storm flows and decrease flooding hazards. In this example from

Golden et al. (2021, used by permission), incorporating the floodwater storage and attenuation functions of non-

floodplain wetlands (NFWs, here) resulted in substantive decreases in flood-stage heights (i.e., modeled stream

outcomes incorporating non-floodplain wetlands reached neither 50 yr nor 100 yr floods extents). The example from

Golden et al. (2021) is of USGS Pipestem Creek gage 06469400, draining approximately 1,800 km².

**6        Global Non-Floodplain Wetlands: Continuing advancements and conclusion**

Noting the challenges in accurately identifying non-floodplain wetlands – including small size, frequent

non-perennial hydrological inundation, soil saturation rather than overlying surface water, and canopy or

cloud cover obstructing satellite or airborne detection – recommendations for advanced analyses of non-

floodplain wetland extent hinge initially on the use of ancillary data sources. For instance, global

assessments will be improved through wall-to-wall high resolution digital elevation models that are used

to identify depressions on the landscape (e.g., Wu et al., 2019b). Though not all landscape depressions are

non-floodplain wetlands (or wetlands at all), analyses that include depressions may find improved



performance when used in combination with vegetation-based assessments or spectral analyses
identifying water (Devries et al., 2017; Evenson et al., 2018b). Similarly, emerging synthetic aperture
radar-based landscape classifications (e.g., Huang et al., 2018; Martinis et al., 2022; Brown et al., 2022)
and both airborne and satellite-borne hyperspectral and advanced analyses, including LiDAR, as well as
analytical capabilities (e.g., machine-learning approaches, object-oriented classifications, Berhane et al.,
2018); topographically based models, Xi et al., 2022) hold great promise for improved resolution and
performance in identifying non-floodplain wetlands (Christensen et al., 2022).

The keys to quantifying the functional contributions, ecosystem services, and watershed-scale resilience
conferred by non-floodplain wetlands through hydrological, biogeochemical, and biological processes are
found through, as a first principle, identifying the spatial extent and configuration of this disappearing and
imperiled aquatic system (Creed et al., 2017; Lane et al., 2022). This novel geospatial dataset, freely
available (https://gaftp.epa.gov/EPADataCommons/ORD/Global_NonFloodplain_Wetlands/, Lane et al.,
2023), provides for sustainable management of an important aquatic resource and advances the global
assessment of non-floodplain wetland functions by facilitating non-floodplain wetland inclusion in both
existing models and those under development (Golden et al., 2021).

**7      Data availability**

The data are available on the United State Environmental Protection Agency's Environmental Dataset
Gateway (DOI: https://doi.org/10.23719/1528331, Lane et al., 2023) or
https://gaftp.epa.gov/EPADataCommons/ORD/Global_NonFloodplain_Wetlands/, (last accessed
12/06/2022). Here, we provide global gridded floodplain (90 m, GFPLain90, ~/Global_Floodplains),
global gridded wetlands (30 m, Global Wetlands, ~/Global_Wetlands), and global gridded non-floodplain
wetlands (30 m, Global NFWs, ~/Global_NFWs) for each of the 3142 HydroBASINS, organized by
HydroBASINS region (see, e.g., Table 7).




**Author contributions.** CL, JC, HG, and ED conceptualized the study, developed the formal analysis, and

conducted and/or assisted the data validation. CL wrote and edited the manuscript, while JC and HG

reviewed and edited the manuscript. ED also developed the methodology, curated the data, conducted the

formal spatial analysis, validated the data, visualized the data, and reviewed and edited the manuscript.

QW and AR assisted in methodology development, validated the study outputs, conducted formal

analyses, and reviewed and edited the manuscript.

750

**Competing interests.** The corresponding authors have declared that none of the authors have any

competing interests.

753

**Disclaimer.** Publisher's note: Copernicus Publications remains neutral with regard to jurisdictional

claims in published maps and institutional affiliations.

756

**Acknowledgements.** We greatly appreciate the scientific contributions and stimulative discussions in the

papers led by Ardalan Tootchi, Sean Woznicki, Fernando Nardi, Oliver Wing, Paul Bates, and their co-

authors that inspired us to complete these analyses. Jeremy Baynes and John Johnston conducted critical

reviews to improve this manuscript, and their efforts are acknowledged. This paper has been reviewed in

accordance with the US Environmental Protection Agency's peer and administrative review policies and

approved for publication. Mention of trade names or commercial products does not constitute

endorsement or recommendation for use. Statements in this publication reflect the authors' professional

views and opinions and should not be construed to represent any determination or policy of the US

Environmental Protection Agency.


**Review statement.** This paper was edited by [Topical Editor] and reviewed by [names/anonymous].




**Appendix A: Abbreviations**


AEB     Aggregate error bias
CCI     Climate change initiative
CONUS     Conterminous United States
CSI     Critical Success Index
CW-WTD     Composite wetland-water table depth
DEM     Digital elevation model
EB     Error Bias
EPA     Environmental Protection Agency
ESA     European Space Agency
FA     False Alarm
FEMA     Federal Emergency Management Agency
GDW     Groundwwater-driven wetlands
GFPlain     Global Floodplain
GIEMS-D15     Global Inundation Extent from Multi-Satellites Downscaled - 15 arcseconds
GIS     Geographic information systems
GLWD     Global Lakes and Wetlands Database
GNFW     Global Non-floodplain wetlands
GSW     Global surface water
GW     Global wetlands
H     Hit Rate
HUC     Hydrologic unit code
IPCC     Intergovernmental Panel on Climate Change
LIDAR     Light detection and ranging
MAE     Mean absolute error
MERIT     Multi-Error Removed Improved Terrain





| 795 | ML | Machine learning |
|---|---|---|
| 796 | NFW | Non-floodplain wetland |
| 797 | NLCD | National Land Cover Database |
| 798 | NWS | National Weather Service |
| 799 | P | Precision |
| 800 | RFW | Regularly flooded wetland |
| 801 | SAR | Synthetic aperture radar |
| 802 | USA | United States of America |
| 803 | USGS | United States Geological Survey |
| 804 | UTM | Universal Transverse Mercator |
| 805 | WTD | Water table depth |




**Appendix B: Supplemental Tables and Figures**

**Table B1.** Descriptive characteristics of the 21 verification basins located throughout the CONUS (see Fig. 2).
Majority Köppen-Geiger classification follows Beck et al. (2018). Climatological data were acquired from the
PRISM Climate Group (Parameter-elevation Regressions on Independent Slopes Model, prism.oregonstate.edu/,
accessed 09/26/2022) using the 30-year annual normals for each watershed. Land use data and descriptions are from
the 2019 NLCD (www.mrlc.gov/data, accessed 09/26/2022) and represent the land use class with the greatest areal
abundance. Average elevation was derived from the USGS National Elevation Dataset (https://www.usgs.gov/3d-
elevation-program, accessed 01/13/2022). Global Wetland Count are the counts of wetlands from the derived Global
Wetland database within each watershed after region-grouping the data using a four-direction contagion criterion
(i.e., pixels immediately adjacent in any of the four cardinal directions are considered part of a unique, multi-pixel
wetland, ArcGIS Pro v.2.9.1, Redlands, California).

| Hydrologic Unit Code ID | Area (km²) | Köppen-Geiger | Mean Annual Temp (ºC) | Mean Annual Rainfall (m) |
|---|---|---|---|---|
| HUC_0101 | 18,906 | Dfb | 4.0 | 1.1 |
| HUC_0103 | 15,287 | Dfb | 5.4 | 1.2 |
| HUC_0106 | 10,800 | Dfb | 7.5 | 1.3 |
| HUC_0203 | 12,490 | Dfa | 11.6 | 1.2 |
| HUC_0208 | 47,449 | Cfa | 13.7 | 1.2 |
| HUC_0304 | 47,899 | Cfa | 16.4 | 1.3 |
| HUC_0313 | 52,169 | Cfa | 18.1 | 1.4 |
| HUC_0501 | 30,371 | Dfb | 8.6 | 1.2 |
| HUC_0706 | 22,257 | Dfa | 8.1 | 1.0 |
| HUC_0804 | 53,108 | Cfa | 17.5 | 1.4 |
| HUC_1003 | 51,431 | BSk | 5.6 | 0.4 |
| HUC_1015 | 37,098 | Dfa | 8.7 | 0.5 |
| HUC_1016 | 54,743 | Dfa | 6.4 | 0.6 |
| HUC_1024 | 35,237 | Dfa | 10.8 | 0.9 |
| HUC_1029 | 48,204 | Dfa | 13.2 | 1.1 |
| HUC_1304 | 48,126 | BSh | 18.6 | 0.4 |
| HUC_1601 | 19,463 | BSk | 5.7 | 0.5 |
| HUC_1708 | 16,101 | Csb | 9.0 | 2.1 |
| HUC_1711 | 35,651 | Csb | 8.2 | 2.0 |
| HUC_1805 | 11,341 | Csb | 14.9 | 0.7 |
| HUC_1808 | 11,789 | BSk | 8.4 | 0.4 |



† Köppen-Geiger Class Descriptions (Beck et al. 2018): BSh (arid, steppe, hot), BSk (arid, steppe, cold), Cfa
(temperate, no dry season, hot summer), Csb, (temperature, dry season, warm summer), Dfa (cold, no dry season,
hot summer), Dfb (cold, no dry season, warm summer)

**Table B1.** (Continued)

| Hydrologic Unit Code | Majority Land Use Coverage | Majority Land Coverage Description | Global Wetland Count | Average Elevation |
|---|---|---|---|---|
| HUC_0101 | 43 | Mixed Forest | 2,141 | 296 |
| HUC_0103 | 43 | Mixed Forest | 2,202 | 300 |
| HUC_0106 | 43 | Mixed Forest | 2,799 | 169 |
| HUC_0203 | 41 | Deciduous Forest | 2,438 | 82 |
| HUC_0208 | 41 | Deciduous Forest | 13,934 | 187 |
| HUC_0304 | 90 | Woody Wetlands | 14,643 | 127 |
| HUC_0313 | 42 | Evergreen Forest | 27,056 | 147 |
| HUC_0501 | 41 | Deciduous Forest | 6,310 | 484 |
| HUC_0706 | 82 | Cultivated Crops | 3,100 | 300 |
| HUC_0804 | 42 | Evergreen Forest | 12,242 | 85 |
| HUC_1003 | 71 | Herbaceous | 11,852 | 1349 |
| HUC_1015 | 71 | Herbaceous | 8,628 | 961 |
| HUC_1016 | 82 | Cultivated Crops | 61,482 | 464 |
| HUC_1024 | 82 | Cultivated Crops | 11,995 | 341 |
| HUC_1029 | 81 | Hay/Pasture | 23,935 | 297 |
| HUC_1304 | 52 | Shrub/Scrub | 1,733 | 995 |
| HUC_1601 | 52 | Shrub/Scrub | 2,642 | 1981 |
| HUC_1708 | 42 | Evergreen Forest | 1,986 | 552 |
| HUC_1711 | 42 | Evergreen Forest | 6,562 | 621 |
| HUC_1805 | 52 | Shrub/Scrub | 1,208 | 222 |
| HUC_1808 | 52 | Shrub/Scrub | 1,089 | 1625 |




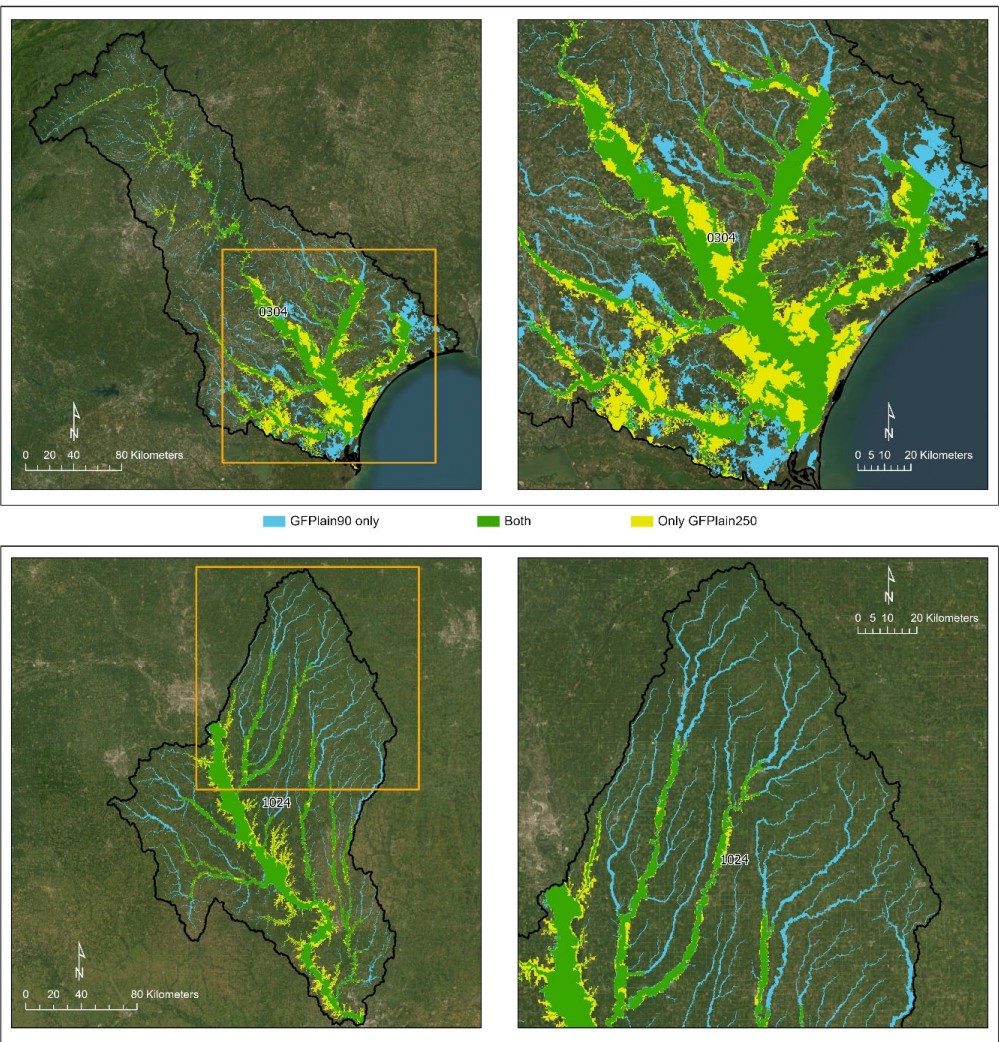

**Figure B1.** Comparison of floodplain extents derived from GFPlain90 (this study) and GFPlain250 (Nardi et al.,

2019). The right-hand panels are the inset area outlined in the orange box on the left panels; the top panels represent

an eastern coastal watershed (HUC_0304) whereas the bottom panels are from a midwestern US watershed

(HUC_1024). The full extent of the riverine network is evident in the GFPlain90 dataset, which was derived from 90

m resolution DEMs in contrast to the 250 m pixel size of the GFPlain250. Satellite imagery sourced from ESRI

(2022).



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
