# Peer review of "Mapping Global Non-Floodplain Wetlands 2 Charles R. Lane1, Ellen D'Amico2, Jay R. Christensen3,★, Heather E. Golden3,★, Qiusheng Wu4, and 3 4 Adnan Rajib5 5 6 1 U.S. Environmental Protection Agency, Office of Research and"

_Earth System Science Data, 2023_

## Author Comment (AC1)

**RC1 – Youjiang Shen**

**General comments**

Lane et al. (2023) presents the initial global geospatial dataset of non-floodplain wetlands by integrating the global floodplain and wetland datasets. The floodplain dataset is produced utilizing an existing algorithm and the MERIT Hydro dataset, whereas the wetland dataset is resampled and modified with greater precision, incorporating previous 500 m CW-WTD data and the GSW and CCI datasets. The authors evaluate the datasets in 21 CONUS watersheds, examining their locations and extents. The study provides valuable insights for hydrological and biogeochemical scientists investigating the impacts and functions of (eco)hydrological cycles. Nonetheless, some aspects require further explanations/modifications in the main text. So, I recommend major revision. Please find my comments in the following.

**Response:** We appreciate the time that Reviewer 1, Youjiang Shen, has taken to comment on the manuscript and look forward to incorporating the comments and criticisms to improve the paper. Thank you.

**Major Comments**

- What led to the authors' selection of Woznicki et al.'s (2019) machine learning-based floodplain dataset for validating their GFPlain90 floodplain? Are there alternative floodplain datasets available for cross-validation?

**Response:** Thank you for this question. We wanted to challenge our GFPlain90 algorithm by acquiring a validation dataset that was of higher resolution (e.g., 30 m), accuracy (~79%), network extent (see, e.g., Figure B1), and freely available relative to other datasets. For instance, the original GFPlain250 (Nardi et al. 2019) data is available at 250 m and the stream network extent is limited versus the Woznicki et al. data layer (see Figure B1). Furthermore, the Woznicki machine learning-based (ML) approach incorporates soil-based, DEM-based (i.e., topographic), and land cover data to map likely floodplain data, implying a more accurate floodplain relative to approaches that use fewer predictors (e.g., floodplains based only on remotely sensed data). The flood extent models of Sampson et al. (2015) are 90 m, meaning that each pixel covers an area 9x greater than that of a single floodplain pixel of Woznicki et al. (2019). The higher-resolution version of those data is not publicly available (Wing et al. 2017). Lastly, within the United States, the floodplain data used to train the Woznicki et al. (2019) ML model is the best available: the Flood Insurance Rate Maps (FIRM). However, these maps only cover about 40% of the conterminous United States (Woznicki et al. 2019). Therefore, we felt the Woznicki et al. (2019) data set, built on the FIRM data layer, was the best available for our purposes (as we detail in L315-326).

Nardi, F., A. Annis, G. Di Baldassarre, E. R. Vivoni and S. Grimaldi (2019). "GFPLAIN250m, a global high-resolution dataset of Earth's floodplains." Scientific Data 6: 180309.

Sampson, C. C., A. M. Smith, P. D. Bates, J. C. Neal, L. Alfieri and J. E. Freer (2015). "A high-resolution global flood hazard model." Water Resources Research 51(9): 7358-7381.

Wing, O. E. J., P. D. Bates, C. C. Sampson, A. M. Smith, K. A. Johnson and T. A. Erickson (2017). "Validation of a 30 m resolution flood hazard model of the conterminous United States." Water Resources Research 53(9): 7968-7986.

Woznicki, S. A., J. Baynes, S. Panlasigui, M. Mehaffey and A. Neale (2019). "Development of a spatially complete floodplain map of the conterminous United States using random forest." Science of The Total Environment 647: 942-953.

- Furthermore, why did the authors limit their watershed-scale comparison of their global wetland product versus the NLCD wetlands to only 21 watersheds? If feasible, it would be more beneficial to provide explicit validations for as many watersheds as feasible to demonstrate the accuracy of the wetland product.

**Response:** Thank you for the observation and comment. The 21 watersheds analyzed cover the breadth of the CONUS as well as a very wide range of climatic, topographic, and land use characteristics (see, e.g., Figure 2 and Table B2). In addition, these watersheds were amongst the best performing in the floodplain analyses of Woznicki et al. (2019). That is, Woznicki et al. (2019) noted an overall accuracy of ~79%; we wanted to ensure that our analyses included top-performing quartile of watersheds while also spanning the gradients noted above to further challenge our floodplain model. In addition, the area covered by our 21 watersheds is 680,000 km$^2$, or roughly the size of France and England combined. These 21 watersheds cover a large area with wetland data of uniform consistency and temporal data acquisition and methodological approaches (e.g., the NLCD is at 30 m resolution, has published performance statistics [Wickham et al. 2021], and incorporates ancillary datasets to improve its accuracy [Jim et al. 2019]). Considering the spatial scale, nearly 700,000 km$^2$ extent, and varied Köppen-Geiger classes of the assessed watersheds we would suggest that the number was a reasonable number to demonstrate the performance of the different products.

Jin, S., C. Homer, L. Yang, P. Danielson, J. Dewitz, C. Li, Z. Zhu, G. Xian and D. Howard (2019). "Overall Methodology Design for the United States National Land Cover Database 2016 Products." Remote Sensing 11(24): 2971.

Wickham, J., S. V. Stehman, D. G. Sorenson, L. Gass and J. A. Dewitz (2021). "Thematic accuracy assessment of the NLCD 2016 land cover for the conterminous United States." Remote Sensing of Environment 257: 112357.

- Why did the authors opt to compare their wetland and non-floodplain wetland products to the NLCD wetlands?

**Response:** We appreciate the question. The NLCD was chosen as it is a publicly available raster dataset with more consistent methodology than, for instance, the National Wetlands Inventory (vector-based, www.fws.gov/wetlands/) data. That said, the NLCD does incorporate ancillary data, such as the NWI, soils data, and topographic position when deriving the wetland classes (Jin et al. 2019). A potential downside to using the NLCD is that the pixel size is 30 m for all data, including wetlands, versus a vector data layer (or a smaller pixel size dataset). However, it is considered the best available land cover data layer for use in the United States giving its consistent classification scheme, high classification accuracy, and extensive validation process.

Jin, S., C. Homer, L. Yang, P. Danielson, J. Dewitz, C. Li, Z. Zhu, G. Xian and D. Howard (2019). "Overall Methodology Design for the United States National Land Cover Database 2016 Products." Remote Sensing 11(24): 2971.

- To better understand the breadth of available regional and global wetland and floodplain datasets, I recommend that the authors include a table summarizing such datasets in their work.

**Response:** Thank you for the recommendation. The scientific community is fortunate to have multiple review papers that address the availability of global wetland and, to a lesser degree, floodplain data. We therefore deliberately chose to not belabor the paper by including tables describing wetland data sets, such as those presented in recent review papers: Shengjie Hu and others (Hu et al. 2017, see their Table 1), Ardalan Tootchi and others (Tootchi et al. 2019, see Table 1), Nick Davidson and others (Davidson et al. (2018, Table S1), and the recently published data from Xiao Zhang and others (Zhang et al. 2023, see Table 1). However, new global land cover datasets that include a proportion of wetlands – ESA WorldCover, ESRI Global Land Cover, and Dynamic World – have not been mentioned in previous papers. In response to this helpful comment, we have therefore included a new Appendix Table B1 in the appendix that introduces these additional data sources as well as pointed readers to the recent papers in the revised manuscript (see L114-117):

> Several recent publications review the available wetland-focused datasets, including Hu et al. (2017, their Table 1), Davidson et al. (2018, their Table S1), Tootchi et al. (2019, their Table 1), and Zhang et al. (2023, their Table 1). We summarize additional emerging global land cover data sets related to surface water and wetlands that were not covered in the above reviews in Appendix Table B1.

**Table B1.** Emerging global land cover datasets related to surface water and wetlands.

| Data Set | Resolution | Years of Data | Wetland Classes | Image Sources | Reference and website |
|---|---|---|---|---|---|
| ESA WorldCover | 10 m | 2020-2021 | Permanent water bodies; herbaceous wetland; mangroves | Sentinel-1 & Sentinel-2 | Zanaga et al. (2021); https://esa-worldcover.org |
| Esri Global Land Cover | 10 m | 2017-2022 | Water; flooded vegetation | Sentinel-2 | Karra et al. (2021) https://livingatlas.arcgis.com/landcover |
| Dynamic World | 10 m | 2015-2023 | Water; flooded vegetation | Sentinel-2 | Brown et al. (2022) https://dynamicworld.app/ |

Davidson, N. C., E. Fluet-Chouinard and C. M. Finlayson (2018). "Global extent and distribution of wetlands: trends and issues." Marine and Freshwater Research 69(4): 620-627.

Hu, S., Z. Niu and Y. Chen (2017). "Global Wetland Datasets: a Review." Wetlands 37(5): 807-817.

Zhang, X., L. Liu, T. Zhao, X. Chen, S. Lin, J. Wang, J. Mi and W. Liu (2022). "GWL_FCS30: global 30 m wetland map with fine classification system using multi-sourced and time-series remote sensing imagery in 2020." Earth Syst. Sci. Data Discuss. 2022: 1-31.

Tootchi, A., A. Jost and A. Ducharne (2019). "Multi-source global wetland maps combining surface water imagery and groundwater constraints." Earth Syst. Sci. Data 11(1): 189-220.

Similarly, we further considered expounding on the availability of global flood models in the revised text, but we did not incorporate these data into a supplemental table due to the level of specificity associated with each model. However, in response to this review comment, we aimed to better highlight the previous literature in the revised manuscript. Hence, we added the following text to L135:

Concurrent with increasingly available global land cover and wetland data, there is an increasing global focus on deriving floodplain and flood hazard-prone areal extents within river networks based on high-resolution topographic data coupled with hydrologic and/or hydraulic modeling (Tullos, 2018; Kundzewicz et al., 2018). The past decade has seen development of multiple regional to continental flood models that span physically based approaches (e.g., 1-,2-, and 3-D hydrodynamic models) to empirical models (including machine-learning approaches and statistical models) (see review by Mudashuru et al. 2021). On the global scale, openly accessible global flood models include those reviewed by Hoch and Trigg (2019), namely CaMa-Flood (Yamazaki et al. 2011), GLOFRIS (Winsemius et al. 2013), JRC (Dottori et al. 2016), CIMA-UNEP (Rudari et al. 2015), Fathom (Sampson et al. 2015), and ECMWF (Papperberger et al. 2012). For instance, Sampson et al. (2015) created a global 90 m map of flood-prone areas between 60°N and 56°S using a regional flood-frequency model. More recently, Nardi et al. (2019) published a global floodplain dataset at 250 m resolution that extended from 60o N to 60o S developed through a geomorphic or terrain-based analyses of floodplain elevations and maximum flood-prone areas based on a drainage-area scaling variable (Rajib et al. 2021). The evolution of the MERIT Hydro 90 m global hydrography dataset by Yamazaki et al. (2019) and machine-learning approaches (e.g., Zhao et al. 2021) has created additional opportunities to further advance the derivation of global floodplains, with improved identification of flow accumulation area, river-basin shape, and river channel location.

- It is recommended that the authors add a section to thoroughly discuss the uncertainties and limitations associated with their proposed dataset. For instance, the authors mentioned that the various input maps capture different types of wetlands (Lines 241-246).

**Response:** We appreciate the comment, and have incorporated additional discussion on the limitations in Section 6 (L739):

The Global NFW dataset is not perfect, yet it incrementally advances the current understanding of the potential extent of this important aquatic resource. Limitations of the global dataset (see also Section 4) include the error-propagation and imperfections of the input data layers, including the relatively coarse nature of four of the main input data layers (i.e., the 1000 m groundwater data from Fan et al. (2013), 500 m CW-WTD from Tootchi et al. (2019), 500 m GIEMS-D15 from Fluet-Chouinard et al. (2015), and the 300 m CCI from Herold et al. (2015)) relative to the target wetland size as clearly evident in Figure 4. We additionally acknowledge that omission and commission errors remain within this global data product. For instance, our floodplain-masking process may have inadvertently misassigned pixels derived at 500 m into either non-floodplain or floodplain groups. Though data were not lost when we resampled downwards to 30 m from 500 m, the topological relationships were not necessarily maintained, adding error to the determination of floodplain or non-floodplain pixel status (especially as it relates to those pixels proximate to floodplains). Though imperfect, we suggest Global NFW data should be cautiously incorporated into hydrological, biogeochemical, and biological models to account for the important functions non-floodplain wetlands perform.

- Additionally, it is worth discussing why the CW-WTD dataset is used as the benchmark dataset for mapping the wetland data, considering that it has a low Pearson correlation value (r=0.34) with other wetland datasets such as GLWD and Hu et al. (2017b). Is it possible to use other input datasets?

**Response:** Thank you for the comment. We considered multiple data sources when we initiated this research in 2020. However, we aligned on using the Tootchi et al. (2019) dataset because of its reported accuracy at global scales, the ready availability of the data, and the inclusion of multiple input types to

successfully capture wetlands maintained by both surface and ground water (e.g., RFWs and GDWs). As we articulate in the concluding section, we envision subsequent researchers to build on and improve from our efforts using additional remotely sensed data and analytical approaches.

- A table that outlines available alternative inputs/benchmark validations for the products proposed in this study could be included.

**Response:** Thank you for the suggestion. We have incorporated additional text regarding the availability of other data layers (see response to question on available regional and global wetland and floodplain datasets, above). We believe that the data validation we conducted using the NLCD was thorough (see responses above to the Reviewer's second and third bullets). One of our main reasons in publishing in ESSD was to ensure that the data was widely available, allowing any and all users to assess the utility of our data for their own purposes using validation data at the appropriate scale for their work.

- There appears to be a lack of clarity regarding the methodology employed for generating the global wetland data.

**Response**: We appreciate the comment and addressed the potential lack of clarity based on the comments below (e.g., revisions to Section 2.2 on Global Wetland data).

- The rationale for not directly utilizing the original 30 m GSW, 300 m CCI, 500 m GIEMS-D15, and 1 km GDW for creating the global wetland data remains unclear.

**Response:** We appreciate the comment. The approach by Tootchi et al. (2019) focused on wetland systems and had wetland-specific reported accuracy and limitations; it was, and to a degree remains, the state of the science with global wetland extents (upon which we are building with these data). That is the primary reason the CW-WTD was used. However, the "all or nothing" decision in developing the CW-WTD required >50% of the pixel to be identified as wetland by Tootchi et al. (2019) as they developed their model. Thus, wetland extents that were below that threshold (as we mentioned in L124-133) would be unmapped. By incorporating input layers from global datasets that included wetlands (such as the CCI) or inundation (e.g., the GSW), we aimed to include more of the smaller wetlands that Tootchi et al. (2019) acknowledged would be missing from their data.

- It is noteworthy that resampling the 500 m CW-WTD product, which was originally based on the 30 m GSW, 300 m CCI, 500 m GIEMS-D15, and 1 km GDW, to 30 m and subsequently incorporating any detected wetland pixels from the resampled 30 m CCI data and inundated pixels from the 30 m GSW data, may introduce discrepancies as compared to directly utilizing the original GSW, CCI, GIEMS-D15, and GDW for wetland mapping.

**Response:** We thank the reviewer for the observation. The incorporation of the original data layers at their original resolution was premediated to overcome an acknowledged deficiency of the Tootchi et al. (2019) data layer, the omission of smaller wetland systems (i.e., those <12.5 ha; see L124+). The improvements in performance metrics when focusing on non-floodplain wetlands as noted in Section 3.2 (see also Table 3 and Table 4, Line 473-483, etc.), we feel, support the approach we have taken.

- Moreover, it is apparent that the current methodology potentially utilizes the information from both GSW and CCI datasets twice, though this may be open to interpretation. Nonetheless, it is acknowledged that this may require further clarification, and any potential misunderstandings on the author's part are sincerely regretted.

**Response:** We appreciate the observation. However, the CW-WTD used the above-mentioned datasets to identify wetlands only when the totality of the input designation of "wetland" (based on the presence of water, as in the GSW, or the classification of wetland, as in the CCI) was sufficient to cover >50% of the 500 m pixel used by Tootchi et al (2019). Our data, as detailed in the flow chart in Figure 1, would then be equivalent to that of Tootchi et al. (2019). The differences would emerge when wetlands (as defined in Section 2.2.2), were outside Tootchi et al.'s 500 m wetland pixels (>50% of the pixel was wetland). In those instances, our methods added wetland pixels as identified by CCI, GSW, etc. to the Global Wetland data layer that were removed by the 50% thresholds.

**Minor comments**

The subscripts in Table 1 require further clarification as they are not clearly understandable.

**Response:** Thank you for the requested clarification. We have added clarifying text to Table 1 noting that the subscript "1" equates to a positive outcome or overlapping extent for either the modeled or benchmark data whereas the zero means no data or a negative outcome. For instance, in the analysis of a watershed, every modeled floodplain pixel that overlaps a benchmark floodplain pixel (i.e., the Woznicki data layer) would have a value of 1 and the added sum of those would represent the value of $M_1B_1$. Modeled floodplain pixels (i.e., GFPlain90) that do not overlap the benchmark would also be given a value of 1 and be similarly summed for the whole watershed; this summed value equates to $M_1B_0$.

In Section 2.4.2, it would be preferable to have a table containing all these equations for the sake of conciseness.

**Response:** We appreciate the observation and revised the paper accordingly, inserting a new table (Table 2, copied below). Thank you.

**Table 2.** Performance metrics used in validation assessments of floodplain and wetland data layers. Data for assessment follow that given in Table 2 and modified from Wing et al. (2017), with the exception of equations 7 and 8 (see text).

| Equation Number | Metrics | Equation | Range |
|---|---|---|---|
| 1 | Hit Rate (H) | $Hit\ Rate\ (H) = \frac{M_1B_1}{M_1B_1 + M_0B_1}$ | 0 – 1, higher is "better" |
| 2 | Precision (P) | $Precision\ (P) = \frac{M_1B_1}{M_1B_1 + M_1B_0}$ | 0 – 1, higher is "better" |
| 3 | False Alarm Ratio (FA) | $False\ Alarm\ Ratio\ (FA) = \frac{M_1B_0}{M_1B_0 + M_1B_1}$ | 0 – 1, lower is "better" |
| 4 | Critical Success Index (CSI) | $Critical\ Success\ Index\ (CSI) = \frac{M_1B_1}{M_1B_1 + M_0B_1 + M_1B_0}$ | 0 – 1, higher is "better" |
| 5 | F1 | $F1 = 2\left(\frac{H \times P}{H + P}\right)$ | 0 – 1, higher is "better" |
| 6 | Error Bias (EB) | $Error\ Bias\ (EB) = \frac{M_1B_0}{M_0B_1}$ | 0 - ∞; <1 underprediction, 1 = no bias, >1 indicate overprediction |
| 7 | Mean Absolute Error (E$_A$) | $Mean\ Absolute\ Error\ (E_A) = \frac{\sum_{i=1}^{N}|M-B|}{N}$ | 0 – 1, lower is "better" |

| 8 | Aggregate Error Bias ($B_A$) | $Aggregate\ Error\ Bias\ (B_A) = \frac{\sum_{i=1}^{N} M-B}{N}$ | -1 to 1, negative values indicate underprediction, positive values overprediction |

Line 358: Is it necessary to include citations for all 7 references to define the Hit Rate? It would be more appropriate to include the most relevant reference for the definition in the article and please balance the number of citations throughout the paper.

**Response:** Thank you for the comment. The number of references associated with the metric has been decreased to three.

Line 189: Please clarify the upscaling parameters.

**Response:** We appreciate the comment and acknowledge the omission in our manuscript. Nardi et al. (2006) analyzed the evolution of different scaling parameters and the resulting sensitivity of the floodplain extent. Following that, in their seminal 2019 paper, Nardi et al. (2019, p. 5) discussed the analyses they conducted on varying parameter sensitivity in their algorithm, concluding that their analyses, "…[support] the use of a constant parameterization at the global scale with $b = 0.30$…" for the power-law exponent. Though not specifically addressed in the 2019 paper, the GFPlain v1.0 algorithm we used includes a default value for the power-law coefficient ($a$) of 0.01. We've added the clarifying information to Line202:

> We then developed the drainage network, drainage area, flow accumulation and flow direction data from these data using the established scaling parameters in Nardi et al. (2019; power-law coefficient ($a$) of 0.01 and dimensionless exponent ($b$) = 0.30).

Nardi, F., E. R. Vivoni and S. Grimaldi (2006). "Investigating a floodplain scaling relation using a hydrogeomorphic delineation method." Water Resources Research **42**(9): 2005WR004155.

Line 581, not clear "global population of wetlands"

**Response:** Thank you for the observation. We have changed this to "Earth's wetlands" for clarity.

Line 663: Please exclude this reference that is still under review.

**Response:** Thank you for catching that. We have updated the citation, as the paper has now been published. We further ensured that no additional 'in review' papers were cited.

Leibowitz, S. G., R. A. Hill, I. F. Creed, J. E. Compton, H. E. Golden, M. H. Weber, M. C. Rains, C. E. Jones, E. H. Lee, J. R. Christensen, R. A. Bellmore and C. R. Lane (2023). "National hydrologic connectivity classification links wetlands with stream water quality." Nature Water 1:370-380. https://doi.org/10.1038/s44221-023-00057-w

Instead of presenting the results in similar tables (tables 2-5), it would be more engaging to include some figures that can get the reader's attention.

**Response:** We appreciate the fresh eyes and helpful suggestion. As the summary values for the floodplain model (was Table 2 in the original submission) are in the text of Section 3.1 and there is a useful figure

(Figure 3) to visually demonstrate performance, we have moved the table to the appendix (now Table B3). As the paper's emphasis is on the wetland data, we maintained the performance metric tables in the main text. However, we created an engaging, we hope, Figure 5 to demonstrate the floodplain abundance across the 1342 global HydroBASINS using our approach.

The statement in lines 261-263 about not losing any data when resampling to a finer resolution needs further clarification.

**Response:** We appreciate the opportunity to clarify, and have made this revision to L279:

> Resampling to a finer resolution (as we did in our analysis) does not result in data losses: the same data are retained but are divided into equal, smaller parts. However, moving from a finer resolution to coarser resolution (as in the CW-WTD dataset's "all-or-nothing" approach) does cause data losses: fine-scale data are necessarily aggregated (often by averaging) to a larger grid cell size, and therefore less information is retained. To compensate for this data loss in the CW-WTD dataset, the finer resolution GSW (30 m) and CCI (300 m) data were added back into the dataset.

The methodology section in 2.2 about the global wetland data could benefit from further clarification. To improve the clarity, it is recommended to introduce the CW-WTD dataset first, followed by the description of the other datasets (RFWs and GDWs). This may help the reader better understand the analysis approach and the role of each dataset in the study.

**Response**: We thank the reviewer and have acted on the suggestion. We believe the section reads more clearly now as we introduced the final product, the CW-WTD and the components (RFW and GDW) in the opening of Section 2.2:

> We specifically used the Tootchi et al. (2019) composite map consisting of both regularly surface-water flooded wetlands ("regularly flooded wetlands," RFWs) and groundwater discharge-maintained wetlands ("groundwater-driven wetlands," GDWs) as the foundation for our global wetland map. Tootchi et al. (2019) merged the RFW and GDW maps, described below, to form a union product with a high correlation with available evaluation data, which they called the composite wetland-water table depth (or CW-WTD).

We then slightly revised the first sentence of the last paragraph:

> Tootchi et al. (2019) created a merged "final" product which that called the composite wetland-water table depth (CW-WTD) map based on the union of the RFW and GDW maps.

**RC2 – Michele Ronco**

This paper discusses the development of a publicly available Global NFW dataset, which provides a foundation for global non-floodplain wetland functional assessments. The dataset reveals that the majority of the globe's wetlands likely occur outside of river floodplains and coastal habitats and is estimated to include over 16 million km2. The dataset will facilitate the inclusion of non-floodplain wetlands in hydrological, biogeochemical, and biological model development, and advance wetland conservation and resource-management goals. Overall, the manuscript is well-written and contains extensive discussion on the procedure followed by the authors. There are two main steps. First the identification of wetlands and floodplain at a global scale, and secondly the derivation of non-floodplain wetlands by subtraction. These

two datasets (i.e. floodplain, and wetlands) had been already introduced in the past and rely on a mix of observations and algorithms. The wetland dataset is resampled from the original 500m to 30m, but there is not a discussion on how this might affect the results (the masking in particular). A second comment is about the temporal coverage of the Global NFW dataset, which I couldn't find in the main text. Then some more concerns are about the second part of the paper on the validation. Table1 is not clear to me, and perhaps could be removed. The metrics (i.e., Eqs. (1)-(8)) could be perhaps moved to the Appendix since are quite standard evaluation scores. Again, there is a resampling involved in order to allow the validation and I would like the authors to discuss how this could impact the metric scores. It should be stressed that the datasets used for validation are not really ground truths since are themselves derived with models. Finally, the authors report quite a significant discrepancy with respect to previous estimates of regional non-floodplain wetlands and I think it would deserve further justification and discussion. Some potential applications of the dataset (e.g., for hydrological model improvements) are mentioned at the end, but I would suggest to also discuss some possible use cases and key breakthroughs that the proposed dataset might be useful for.

**Response:** We greatly appreciate the attention to detail and comments from Michele Ronco, Reviewer 2, and look forward to improving the paper based on the useful and actionable comments received. Thank you for your efforts. We directly respond to the comments below.

The wetland dataset is resampled from the original 500m to 30m, but there is not a discussion on how this might affect the results (the masking in particular).

**Response:** Thank you for the comment. We very deliberately and thoughtfully considered the appropriate resolution for the data. Our main input data layer, the CW-WTD, was developed by Tootchi et al. (2019) using a 500 m cell size that the researchers acknowledged would result in smaller wetland systems being lost (see L124-133). The authors also noted that their "all or nothing" approach to determining wetland presence in a cell would also affect their product's ability to identify smaller wetlands, those <12.5 ha. As Cohen et al. (2016) noted that most non-floodplain wetlands are much smaller (i.e., 2.1 ha, see L130), we resolved to use 30 m to try to identify the target system, non-floodplain wetlands.

As we noted in L292, "…we overlaid our GFPlain90 floodplain data with our mapped Global Wetlands data to mask wetland pixels collocated on the floodplain." The GFPlain90 data were resampled to 30 m, and the Global Wetlands data were also resampled to 30 m. We stated in L279:

> Resampling to a finer resolution (as we did in our analysis) does not result in data losses: the same data are retained but are divided into equal, smaller parts. However, moving from a finer resolution to coarser resolution (as in the CW-WTD dataset's "all-or-nothing" approach) does cause data losses: fine-scale data are necessarily aggregated (often by averaging) to a larger grid cell size, and therefore less information is retained. To compensate for this data loss in the CW-WTD dataset, the finer resolution GSW (30 m) and CCI (300 m) data were added back into the dataset.

However, the reviewer identifies a topic on which we focused a great deal of discussion: when to identify a given wetland pixel as overlapping a floodplain. For instance, if a pixel, originally at 500 m cell size, is overlapped by a floodplain such that 33% of that pixel is covered, does that necessitate marking the entirety of the pixel as a "floodplain wetland"? What if that overlapping percent is 67%? 90%? In the end, we opted to take advantage of the resampling process and simply exclude the 30 m pixels that overlapped rather than developing a cumbersome (at the global scale) if-then procedure regarding masking. We acknowledge that this may result in an undercounting of floodplain wetland pixels, and similarly may result in an over-estimate of non-floodplain wetlands (as wetlands connected to floodplains are

inappropriately split into erroneously termed non-floodplain portions). We acknowledge this by adding the following text to L744 in the revised manuscript:

> We additionally acknowledge that omission and commission errors remain within this global data product. For instance, our floodplain-masking process may have inadvertently misassigned pixels derived at 500 m into either non-floodplain or floodplain groups. Though data were not lost when we resampled downwards to 30 m from 500 m, the topological relationships were not necessarily maintained, adding error to the determination of floodplain or non-floodplain pixel status (especially as it relates to those pixels proximate to floodplains).

A second comment is about the temporal coverage of the Global NFW dataset, which I couldn't find in the main text.

**Response:** We appreciate the insight and acknowledge our omission. We added this information to the abstract for clarity (L35) and addressed this recommendation in L754:

> Similarly, though this Global NFW is a static data layer, land use, development, and climate changes continue to affect the prevalence of wetlands worldwide. Fluet-Chouinard et al. (2023) recently noted a global wetland loss of 21% since 1700, with rapid increases from 1950s onwards. Returning to the identification of wetlands and their spatial location vis-à-vis floodplains, using the preponderance of higher-resolution (i.e., < 30 m) and high-return interval sensors will improve both the spatial and temporal accuracy of these data decreasing commission and omission errors (e.g., Table 8) while increasing the accurate identification of smaller aquatic features that occasional cease to hold standing water.

Then some more concerns are about the second part of the paper on the validation.

**Response:** Thank you for the comment We look forward to clearly addressing the concerns noted regarding the validation process.

Table1 is not clear to me, and perhaps could be removed.

**Response:** We appreciate the comment and acknowledge that Table 1 is slightly unusual in that it contains contextual information rather than quantified data. However, the subsequent use of those categorical classes in the metrics suggests that maintaining this table is useful for readers, especially those unfamiliar with classification metrics. Note that we also further clarified the subscripts in the table, which we hope creates a more useful product for the readers.

The metrics (i.e., Eqs. (1)-(8)) could be perhaps moved to the Appendix since are quite standard evaluation scores.

**Response:** Thank you for the observation and recommendation. The rationale behind including both the abundance of references (see the comment regarding L358 by Reviewer #1) and specifying the metrics using the contextual data given in Table 1 is that there are many different types of metrics, and they have various names. For instance, we use the term "Hit Rate" to Bates and Wing (e.g., Sampson et al. 2015; Wing et al. 2017). However, others have used similar metrics but with different names (e.g., Sangwan and Merwade 2015, Woznicki et al. 2019). The Excel table we crafted to crosswalk the various metrics was expansive. Thus, though we acknowledge they may be standard in some cases, we respectfully note that others may see that differently and therefore chose to maintain the metrics in the main text to target the widest reader audience. However, for clarity and balance we have reduced the number of citations associated with Hit Rate and incorporated the metrics in Table 2 (and out of the main text).

Again, there is a resampling involved in order to allow the validation and I would like the authors to discuss how this could impact the metric scores.

**Response:** We appreciate this comment and have addressed it above in our second response to Review 2.

It should be stressed that the datasets used for validation are not really ground truths since are themselves derived with models.

**Response:** Thank you for the comment. We appreciate and acknowledge that the data used in our assessment of model performance (i.e., Woznicki et al. 2019 and Dewitz 2019) are themselves model based. In some cases, the Woznicki and Dewitz models and data were in fact ground-truthed (e.g., U.S. Federal Emergency Management Agency or National Wetlands Inventory data affirmed the locational and typological accuracy of the modeled data) and accuracy measures were quantified. However, we believe readers of ESSD will acknowledge and understand that the modeled data used in our performance measures have inherent limitations. To that end, we provide performance measures for both the floodplain data (L322+, ~79% correctly identified) and wetland data (L310, approximately 86% of the land covers correctly identified).

Finally, the authors report quite a significant discrepancy with respect to previous estimates of regional non-floodplain wetlands and I think it would deserve further justification and discussion.

**Response:** We appreciate the observation and comment. We believe the comment is addressing the data presented in Table 7 where we show that our Global NFW as well as the Tootchi et al. (2019) CW-WTD data overestimate the abundance of non-floodplain wetlands identified by the NLCD.

We build on the Reviewer's (temporal-focused) previous comment in the revised manuscript (adding to L754):

> Similarly, though this Global NFW is a static data layer, land use, development, and climate changes continue to affect the prevalence of wetlands worldwide. Fluet-Chouinard et al. (2023) recently noted a global wetland loss of 21% since 1700, with rapid increases from 1950s onwards. Returning to the identification of wetlands and their spatial location vis-à-vis floodplains, using the preponderance of higher-resolution (i.e., < 30 m) and high-return interval sensors will improve both the spatial and temporal accuracy of these data, decreasing commission errors (e.g., Table 8) while increasing the accurate identification of smaller aquatic features that occasional cease to hold standing water.

We additionally added the following text to L621:

> This suggests that our first approximation of global non-floodplain wetland estimates may be high, primarily due to the resolution of the input data layers. However, as we discuss below, additional factors than just resolution are likely at play.

In the paper, we discuss the limitations of the combined Wetland NFW data set, acknowledging errors occur from wetlands being obscured by trees (L630), the temporal nature of wetland hydrology resulting in errors of omission (L633-643), and that soil saturation is a critical factor affecting wetland classification (e.g., L643-646).

Further, in developing this paper, we choose not to contrast our study with coarser-resolution data (e.g., the Global Lakes and Wetlands Database) because we felt it could inappropriately "improve" our results. Using the higher-resolution NLCD almost guaranteed higher omission and commission errors. Yet we

used the NLCD because we wanted to demonstrate and acknowledge that improved data and analyses are needed to further advance our work. Just as we built on work by Tootchi, Lehner and Döll, Bates, Wing, Woznicki, Wu, Hu, and others to report a "reasonable first approximation of the total non-floodplain wetland extent" (L593), it is our hope that others will improve upon our work here.

To that end, we suggest that efforts to improve wetland mapping should prioritize the utilization of datasets with higher temporal and spatial resolutions, in addition to leveraging the capabilities of machine learning techniques to enhance the accuracy of global non-floodplain wetland identification (see, e.g., L715-770).

Some potential applications of the dataset (e.g., for hydrological model improvements) are mentioned at the end, but I would suggest to also discuss some possible use cases and key breakthroughs that the proposed dataset might be useful for.

**Response**: We appreciate the comment. The incorporation of non-floodplain wetlands into hydrological, biogeochemical, and biological models is still – unfortunately – in its infancy. In developing the paper, we included two specific examples of the implications of incorporating non-floodplain wetlands into watershed-scale analyses (e.g., Rajib et al. 2020, and Golden et al. 2021), demonstrating that non-floodplain wetlands substantially attenuate storm flows. We further state in L704 that: "…ignoring non-floodplain wetlands in the model resulted in projected critical flood-stage return intervals (e.g., 50 yr and 100 yr floods) being reached within a given modeled time frame…" In other words, fairly significant model errors were the end result in this use-case study (albeit using a different dataset of non-floodplain wetlands than in this paper).

We conclude the Implications section (Section 5) with the adage that the multiple functions of non-floodplain wetlands, "…demand an effective accounting of their spatial extent and configuration, as demonstrated in this novel global dataset." Thus, we feel that the current abundance of examples, including the figured borrowed from Golden et al. (2021) showing the importance of non-floodplain wetlands, serves as a valuable demonstration of user cases and breakthroughs.

---

## Author Response (AR2)

Reviewer #1, Report #2

**Comment**: If authors utilize codes to automatically complete the validation process for the data, I recommend that they validate all watersheds instead of just 21 selected watersheds. It would be beneficial if the author includes the evaluation reports either as an attachment or within the provided dataset. Additionally, if the authors used the codes for validation, it would be good for them to make it publicly accessible for future use by users and for local validation purposes. If the validation part relies on manual efforts, please disregard the previous suggestions. I highly appreciate the authors' revisions, as they have effectively addressed the majority of my concerns.

**Response**: The reviewer makes an excellent point. However, we did not use code to validate the process but instead manually interrogated the spatial data and calculated the performance metrics in a spreadsheet prior to reporting the metrics (e.g., Table 3). Creating publicly available code would greatly speed up the process and allow for a greater number of watersheds to be assessed; we will consider that high on our to-do list in subsequent research.